# Local Function Complexity for Active Learning via Mixture of Gaussian Processes

## Abstract

Inhomogeneities in real-world data, e.g., due to changes in the observation noise level or variations in the structural complexity of the source function, pose a unique set of challenges for statistical inference. Accounting for them can greatly improve predictive power when physical resources or computation time is limited. In this paper, we draw on recent theoretical results on the estimation of *local function complexity* (LFC), derived from the domain of *local polynomial smoothing* (LPS), to establish a notion of local structural complexity, which is used to develop a model-agnostic active learning framework. Due to its reliance on pointwise estimates, the LPS model class is not robust and scalable with respect to large input space dimensions that typically come along with real-world problems. Here, we propose a GPR-based estimate of LFC, which is able to manage the *curse of dimensionality*. To this end, we train a *mixture of experts* (MoE) model where the *experts* are GPR models at different bandwidths. Being the key ingredient in the calculation of LFC, we then estimate *locally optimal kernel bandwidths* as the weighted average of these bandwidth candidates, where the weights are taken from the learned *gate* of the MoE model. We assess the effectiveness of our LFC estimate in an active learning application on a prototypical low-dimensional synthetic dataset, before taking on the challenging real-world task of reconstructing a quantum chemical force field for a small organic molecule and demonstrating state-of-the-art performance at a lower rate of sampling.

## 1 Introduction

Inference problems from real-world data often exhibit inhomogeneities, e.g., the noise level, the density of the data distribution or the complexity of the target function may change over the input space. There exist different approaches from various domains that treat specific kinds of inhomogeneities. For example, Kersting et al. (2007); Cawley et al. (2006) deal with heteroscedasticity by modelling a *local noise variance* function that then improves prediction performance by adapting the model regularization locally. Some approaches adjust bandwidths locally with respect to the input density Wang & Wang (2007); Mackenzie & Tieu (2004); Moody & Darken (1989); Benoudjit et al. (2002). Inhomogeneous complexity can also be tackled by stacking several kernel-linear models with different bandwidths, either learned jointly Zheng et al. (2006); Guigue et al. (2005) or hierarchically Ferrari et al. (2010); Bellocchio et al. (2012). The most widely applicable approaches treat all types of aforementioned inhomogeneities in a unified way (Tresp, 2001; Panknin et al., 2021), which is the path we will pursue in this work. In particular, we will focus on the theory of inhomogeneous complexity under the assumption of homoscedastic noise. In addition, we will apply our derived framework in a heteroscedastic setting to demonstrate practical implications.

The identified inhomogeneities can shed light on the informativeness of certain locations of the input space, which subsequently can be used to guide the sampling process while training—also known as *active learning*. Active learning (Kiefer, 1959; MacKay, 1992; Seung et al., 1992; Seo et al., 2000) is a powerful tool to enhance the performance of a model for inference, whenever the acquisition of labeled training data is expensive. It has been successfully implemented in various regression applications like reinforcement learning (Teytaud et al., 2007), wind speed forecasting (Douak et al., 2013) and optimal control (Wu et al., 2020). Specifically in pharmaceutic applications Warmuth et al. (2003) and in the domain of quantum chemistry (Gubaev et al.

(2018); Tang & de Jong (2019); Huang & von Lilienfeld (2020)) we can benefit from active learning, since the labels stem from computationally expensive first-principles calculations (Chmiela et al., 2017) or even laboratory experiments.

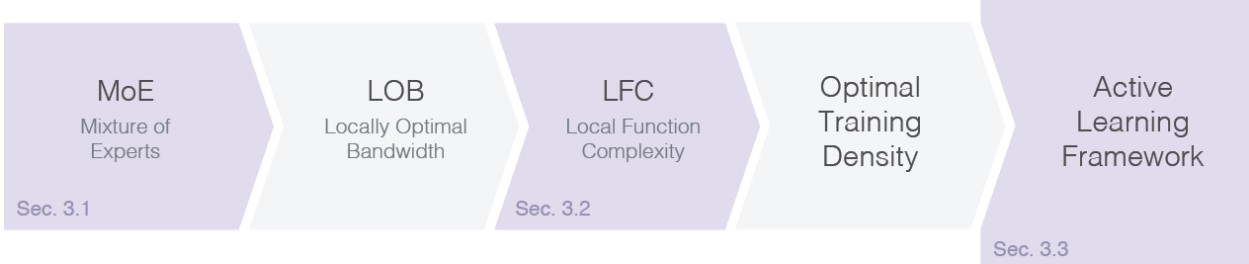

Figure 1: An overview of the steps of our contribution and how they are interlinked. We will elaborate on the main steps in the specified sections.

Our work aims at improving the efficiency of the inference task by identifying local inhomogeneities in the data and exercising this insight to construct better models and training datasets. Specifically, we contribute in three ways:

First, we propose a *sparse mixture of Gaussian processes* model, which extends a single-scale *Gaussian process regression* (GPR)-based model to a multi-scale approach in a natural way, raising the prediction performance whenever the data features multi-scale behavior. In particular, we will set up a *mixture of experts* (MoE) model (see Jacobs et al. (1991); Jordan & Jacobs (1994); Pawelzik et al. (1996)), where each experts is a GPR model that holds an individual, fixed bandwidth. MoE have been applied across the board in the *machine learning* (ML) literature of which an introduction and summary can be found in Yuksel et al. (2012). After training, the latent *gate* model selects the expert that is best suited for the prediction of a test input $x \in \mathcal{X}$.

Second, we propose an estimate to *locally optimal kernel bandwidths* (LOB) from the learned gate of our MoE model, based on which we derive an estimate to the *local function complexity* (LFC) under homoscedasticity— a property that describes the local structural complexity of the regression function. Essentially, LFC is the reciprocal determinant of LOB, calibrated for the local effects of the training input density and noise level, and was originally defined by Panknin et al. (2021) for the *local polynomial smoothing* (LPS) model class (Cleveland & Devlin, 1988). We derive the GPR-based analog by following their definition, but using asymptotic results on the scaling of optimal bandwidths for GPR by Van der Vaart et al. (2007; 2009). In contrast to the LPS-based estimate, our GPR-based version is scalable with respect to the input space dimension. As a scalar-valued function, LFC allows for an easy inspection to gain insights into the local structural complexity of the target function—even for high-dimensional input spaces—making it a problem intrinsic, *interpretable* property of the regression problem. Furthermore, LFC can be used to make models more parsimonious, for example, by placing additional basis functions of a kernel method in more complex regions while removing basis functions in simpler regions of the input space. We will discuss both aspects, interpretability and model parsimony, in the experiments.

Finally, we implement a GPR-based version of the recent active learning framework by Panknin et al. (2021). As opposed to several active learning approaches—where commonly training data is refined bottom-up in a more or less greedy manner Kiefer (1959); MacKay (1992); Seung et al. (1992); Seo et al. (2000); Roy & McCallum (2001); He (2010); Douak et al. (2013); Bull et al. (2013); Gubaev et al. (2018); Goetz et al. (2018)—the fundamental idea of the considered approach is to analyze the distribution of the hypothetically optimal training set in the asymptotic limit of the sample size. Assuming that this limiting distribution exists, we would like to sample training data in a top-down manner from this very distribution, knowing that with growing sample size the training set will eventually become optimal. Panknin et al. (2021) have shown for the *local polynomial smoothing* (LPS) model class that this asymptotic distribution exists, whose density furthermore factorizes into contributions of the test density, heteroscedastic noise and LFC. Given a small, but sufficient training set, these factors can be estimated, allowing the construction of the optimal training density and subsequently enabling the refinement of the training data towards asymptotic optimality.

Here, our contribution is to replace the LPS-based LFC estimate with our GPR-based LFC estimate in the construction of the optimal training density. In doing so, we make this active learning approach suited for problems of higher input space dimensions, which hitherto was missing. It is particularly the almost assumption-free nature of LPS that made the results of Panknin et al. (2021) model-agnostic. In particular, they have demonstrated a superior performance of their approach under reasonable model change. To that effect, we base our results on the nonparametric GPR model with the goal to preserve this property. Hence, we obtain model-agnostic estimates to LFC and a *superior training density*, which we can inspect to gain deeper insights into the MD simulation problem.

In the domain of quantum chemistry, regression problems commonly exhibit multi-scale behavior due to the complex electronic interactions that give rise to any observable property of interest, like the total energy or atomic forces of a system (Bereau et al., 2018; Yao et al., 2018; Grisafi & Ceriotti, 2019; Ko et al., 2021; Unke et al., 2021a). This is why we choose a *molecular dynamics* (MD) simulation experiment to demonstrate the capabilities of our framework in practice. Here, we will apply the sGDML model by Chmiela et al. (2019) as experts, which—besides taking derivative information and invariances of the molecule into account—directly corresponds to a GPR model.

We begin with a formal definition of the considered regression problem and the asymptotic active learning task, and review asymptotic results for LPS and GPR in Sec. 2. In Sec. 3, we describe our MoE model and derive the GPR-based LFC and *superior training density* estimates. We further discuss related work in Sec. 4. In experiments in Sec. 5, we will first demonstrate our MoE approach and the proposed estimates to LFC and the *superior training density* and compare to related work on toy-data. After that, we will apply our framework to a 27-dimensional MD simulation dataset. Finally, we conclude in Sec. 6.

## 2 Preliminaries

We begin with a formal definition of the regression task, the active learning objective and LOB, and a short review of the asymptotic results on LFC and the optimal training distribution of the LPS model in Sec. 2.1 and 2.2, followed by asymptotic results on the optimal bandwidth of GPR in Sec. 2.3.

In the following, we denote by $\boldsymbol{diag}(z) \in \mathbb{R}^{d \times d}$ the diagonal matrix with the entries of the vector $z \in \mathbb{R}^d$ on its diagonal and by $\mathcal{I}_d = \boldsymbol{diag}(\mathbb{1}_d)$ the identity matrix, where $\mathbb{1}_d$ is the vector of ones in $\mathbb{R}^d$.

### 2.1 Formal definition of the regression task and active learning objective

Let $f$ be the target regression function defined on an input space $\mathcal{X} \subset \mathbb{R}^d$ that we want to infer from noisy observations $y_i = f(x_i) + \varepsilon_i$, where $x_i \in \mathcal{X}$ are the training inputs and $\varepsilon_i$ is independently drawn from a distribution with mean $\mathbb{E}[\varepsilon_i] = 0$ and local noise variance $\mathbb{V}[\varepsilon_i] = v(x_i)$. We denote a training set by $(\boldsymbol{X}_n, \boldsymbol{Y}_n)$, where $\boldsymbol{X}_n = (x_1, \ldots, x_n) \in \mathcal{X}^n$ and $\boldsymbol{Y}_n = (y_1, \ldots, y_n) \in \mathbb{R}^n$. For a given model $\widehat{f}$ and a training set $(\boldsymbol{X}_n, \boldsymbol{Y}_n)$, we may define the pointwise *conditional mean squared error* of $\widehat{f}$ in $x \in \mathcal{X}$ by

$$\text{MSE}\left(x, \widehat{f}|\boldsymbol{X}_n\right) = \mathbb{E}_\varepsilon\left[(\widehat{f}(x) - f(x))^2|\boldsymbol{X}_n\right]. \tag{1}$$

Although the model $\widehat{f}$ depends on the training labels $\boldsymbol{Y}_n$, the conditional MSE of $\widehat{f}$ does not, by construction. Given a test probability density $q \in \mathcal{C}^0\left(\mathcal{X}, \mathbb{R}_+\right)$ such that $\int_{\mathcal{X}} q(x)dx = 1$, the *conditional mean integrated squared error* of the model under the given training set is then defined as

$$\text{MISE}\left(q, \widehat{f}|\boldsymbol{X}_n\right) = \int_{\mathcal{X}} \text{MSE}\left(x, \widehat{f}|\boldsymbol{X}_n\right) q(x)dx. \tag{2}$$

With these preparations, the active learning task is to construct a training set $(\boldsymbol{X}_n', \boldsymbol{Y}_n')$ such that

$$\boldsymbol{X}_n' \approx \boldsymbol{\arg\min}_{\boldsymbol{X}_n \in \mathcal{X}^n} \text{MISE}\left(q, \widehat{f}|\boldsymbol{X}_n\right). \tag{3}$$

## 2.2 Locally optimal bandwidths, function complexity and optimal training

Let us consider a family $\widehat{f}^\Sigma$ of *kernel machines* which is characterized by a positive definite bandwidth matrix parameter $\Sigma \in \mathbb{S}^d_{++}$ of a *radial basis function* (RBF) kernel $k^\Sigma(x, x') := |\Sigma|^{-1}\phi(\|\Sigma^{-1}(x - x')\|)$ for a monotonically decreasing function $\phi: \mathbb{R}_+ \to \mathbb{R}_+$. The well known Gaussian kernel is for example implemented by $\phi(z) = \exp\{-\frac{1}{2}z^2\}$.

Given a bandwidth space $\mathcal{S} \subseteq \mathbb{S}^d_{++}$ we define the *locally optimal bandwidth* (LOB) function

$$\Sigma^n(x) = \mathbf{arg\,min}_{\Sigma \in \mathcal{S}} \mathrm{MSE}\left(x, \widehat{f}^\Sigma | \boldsymbol{X}_n\right), \tag{4}$$

assuming that these pointwise minima uniquely exist.

Denote by $m_Q^\Sigma$ the predictor of the LPS model of order $Q$ under bandwidth $\Sigma$ and by $\Sigma_Q^n$ the LOB function (4) of LPS, if it is well-defined. This is the case, e.g., for the *isotropic* bandwidths space $\mathcal{S} = \{\sigma \mathcal{I}_d \mid \sigma > 0\}$ under mild assumptions[1], where we particularly can write $\Sigma_Q^n(x) = \sigma_Q^n(x)\mathcal{I}_d$. For the optimal predictor

$$\widehat{f}^Q_{\mathrm{LPS}} = m_Q^{\Sigma_Q^n(x)}(x) \tag{5}$$

of LPS, letting $\widehat{f} = \widehat{f}^Q_{\mathrm{LPS}}$ in Eq. (3), Panknin et al. (2021) have shown that there exists an optimal training density $p_{\mathrm{Opt}}^{Q,n}$ such that the optimal training set choice in Eq. (3) can asymptotically be obtained by sampling independently and identically $\boldsymbol{X}'_n \sim p_{\mathrm{Opt}}^{Q,n}$. This density exhibits a closed-form

$$p_{\mathrm{Opt}}^{Q,n}(x) \propto \left[\mathfrak{C}_Q^n(x)q(x)\right]^{\frac{2(Q+1)+d}{4(Q+1)+d}} v(x)^{\frac{2(Q+1)}{4(Q+1)+d}}(1 + o(1)), \tag{6}$$

where for an arbitrary training dataset $(\boldsymbol{X}_n, \boldsymbol{Y}_n)$ with $\boldsymbol{X}_n \sim p$,

$$\mathfrak{C}_Q^n(x) = \left[\frac{v(x)}{p(x)n}\right]^{\frac{d}{2(Q+1)+d}} \left|\Sigma_Q^n(x)\right|^{-1} = \left[\frac{v(x)}{p(x)n}\right]^{\frac{d}{2(Q+1)+d}} \sigma_Q^n(x)^{-d} \tag{7}$$

solely depends on the behavior of $f$ in the vicinity of $x$: It scales with the local variation of $f$ and therefore serves as a measure of LFC in $x$. Indeed, for odd $Q$, it is $\mathfrak{C}_Q^n = \mathfrak{C}_Q^\infty(1 + o(1))$, where

$$\mathfrak{C}_Q^\infty(x) = \mathrm{b}_Q\left[x, \mathcal{I}_d\right]^{\frac{2d}{2(Q+1)+d}} \tag{8}$$

for the asymptotic leading bias-term $\mathrm{b}_Q\left[x, \mathcal{I}_d\right]$ of order $(Q + 1)$ (see Eq. (25)), which is a function of the $(Q+1)-th$ derivative $D_f^{Q+1}(x)$ of the target function $f$. For example, $\mathrm{b}_1\left[x, \mathcal{I}_d\right] \propto \boldsymbol{trace}(D_f^2(x))$ is a function of the trace of the Hessian of $f$ (Fan et al., 1997).

The density $p_{\mathrm{Opt}}^{Q,n}$ reflects the need for more training data where the problem is locally more complex or noisy, or where test instances are more likely. In practice, we obtain $\boldsymbol{X}'_n \sim p_{\mathrm{Opt}}^{Q,n}$ by estimating Eq. (6) and (7) from $(\boldsymbol{X}_{n'}, \boldsymbol{Y}_{n'})$ with $\boldsymbol{X}_{n'} \sim p$ for an arbitrary training density $p$, where $n' < n$, followed by adding the remaining $n - n'$ inputs appropriately (see Sec. 3.3).

We can further define a generalized LFC of LPS for $f \in \mathcal{C}^\alpha(\mathcal{X}, \mathbb{R})$, given by

$$\mathfrak{C}_{Q,\mathbb{S}^d_{++}}^n(x) = \left[\frac{v(x)}{p(x)n}\right]^{\frac{d}{2\alpha+d}} \left|\Sigma_{Q,\mathbb{S}^d_{++}}^n(x)\right|^{-1}, \tag{9}$$

which is based on the non-isotropic LOB function $\Sigma_{Q,\mathbb{S}^d_{++}}^n(x) \in \mathbb{S}^d_{++}$—that is, Eq. (4) for the bandwidth space $\mathcal{S} = \mathbb{S}^d_{++}$. Panknin et al. (2021) have analyzed that $\mathfrak{C}_{Q,\mathbb{S}^d_{++}}^n$ will asymptotically be invariant with respect to $v$, $p$ and $n$, such that it is solely a function of the target function $f$ even though it exhibits no explicit solution in terms of $f$, as was obtained in the isotropic case in Eq. (8).

---

[1]For LOB being well-defined in the isotropic case, we generally require a non-vanishing bias and variance in terms of a bias-variance-decomposition of the MSE of the predictor in x, for all $x \in \mathcal{X}$. See, e.g., Eq. (23) for the LPS predictor $m_Q$, or Silverman (1986); Wand & Jones (1994) in more general.

In the light of the generalized LFC, we obtain the *superior training density* as the straight-forward extension of the optimal training density (6) by

$$p_{\text{Sup}}^{Q,\mathbb{S}_{++}^d,n}(x) \propto \left[ \mathfrak{C}_{Q,\mathbb{S}_{++}^d}^n(x) q(x) \right]^{\frac{2\alpha+d}{4\alpha+d}} v(x)^{\frac{2\alpha}{4\alpha+d}} (1 + o(1)). \tag{10}$$

For even $\alpha \in \mathbb{N}$, (10) generalizes (6) with $Q = \alpha - 1$, which proved exact optimality. Now that $\alpha \in \mathbb{R}$ deviates at most by one from the next even integer, we expect that the true optimal training density for general $\alpha$ will not deviate by a lot from $p_{\text{Sup}}^{Q,\mathbb{S}_{++}^d,n}$. The results to LFC and the *superior training density* of LPS indicate their problem intrinsic nature, as they reflect no direct dependence on the LPS model. Accordingly, Panknin et al. (2021) claimed their model-agnostic character, for which they also provided empirical evidence: They observed superior performance when training a regression forest model and an RBF-network under their AL framework. For more details on the LPS model and asymptotic results, we refer to Appendix A.

The construction of $p_{\text{Opt}}^{Q,n}$ crucially depends on reliable estimates to LOB as the key ingredient for the estimation of LFC. While Panknin et al. (2021) provided such an estimate based on Lepski's method (Lepski, 1991; Lepski & Spokoiny, 1997), it does not scale well with increasing input space dimension $d$. This is due to the fact that this pointwise estimate suffers from the *curse of dimensionality* regarding robustness and computational feasibility. The goal of this work is to implement the above active learning framework, but based on a functional LOB estimate in the domain of GPR instead of LPS, since the GPR model class can naturally deal with high input space dimensions (Williams & Rasmussen, 1996). Here, we rely on the model-agnosticity of LFC and the *superior training density*: We expect that LOB estimates based on GPR can be exchanged for LOB estimates based on LPS in the active learning framework, when matching the degree $Q$ to the smoothness of the regression function appropriately.

### 2.3   On the scaling of GPR bandwidths

The major difference between LPS and GPR is, that we keep a fixed model complexity—in the sense of number of basis functions—in the former while there is varying model complexity in the latter as we add further training instances. I.e. for the Gaussian kernel there is an infinite growth of model complexity. When the regularity of the kernel and the target function $f$ match, then, as soon as the training size $n$ becomes large enough, there is no further shrinkage of bandwidth necessary anymore to exactly reproduce $f$. In particular, given enough samples, there is no need for local bandwidth adaption.

However, there is a mismatch if $f \in \mathcal{C}^\alpha(\mathcal{X}, \mathbb{R})$ is $\alpha$-times continuously differentiable since the Gaussian kernel is infinitely often continuously differentiable. As shown by Van der Vaart et al. (2007; 2009), in order to obtain optimal *minimax*-convergence of the predictor (except for logarithmic factors), the associated (global) bandwidth has to follow the asymptotic law

$$\Sigma_{\text{GPR}}^n \propto n^{-\frac{1}{2\alpha+d}}. \tag{11}$$

Note that for $f \in \mathcal{C}^\alpha(\mathcal{X}, \mathbb{R})$, where the theoretical results of LPS apply, the scaling factor $n^{-\frac{1}{2\alpha+d}}$ of LOB in sample size matches exactly for both classes, LPS and GPR. In our work, we will use this analogy to deduce a GPR-based LFC estimate in the same way it was done by Panknin et al. (2021) for LPS.

## 3   Estimating locally optimal bandwidths via mixture of Gaussian processes

In this section, we derive our main contribution, namely the GPR-based active learning framework, which we summarized in Fig. 2. We first describe our GPR-based MoE model in Sec. 3.1 (Fig. 2, A), of which the gate function will be used to derive our GPR-based *superior training density* estimate in Sec. 3.2 (Fig. 2, B). Combining this estimate with the active learning framework from Panknin et al. (2021), we obtain a GPR-based sampling scheme in Sec. 3.3 (Fig. 2, C), which is model-agnostic, asymptotically superior to *random test sampling* and interpretable while being applicable to problems with high input space dimensions.

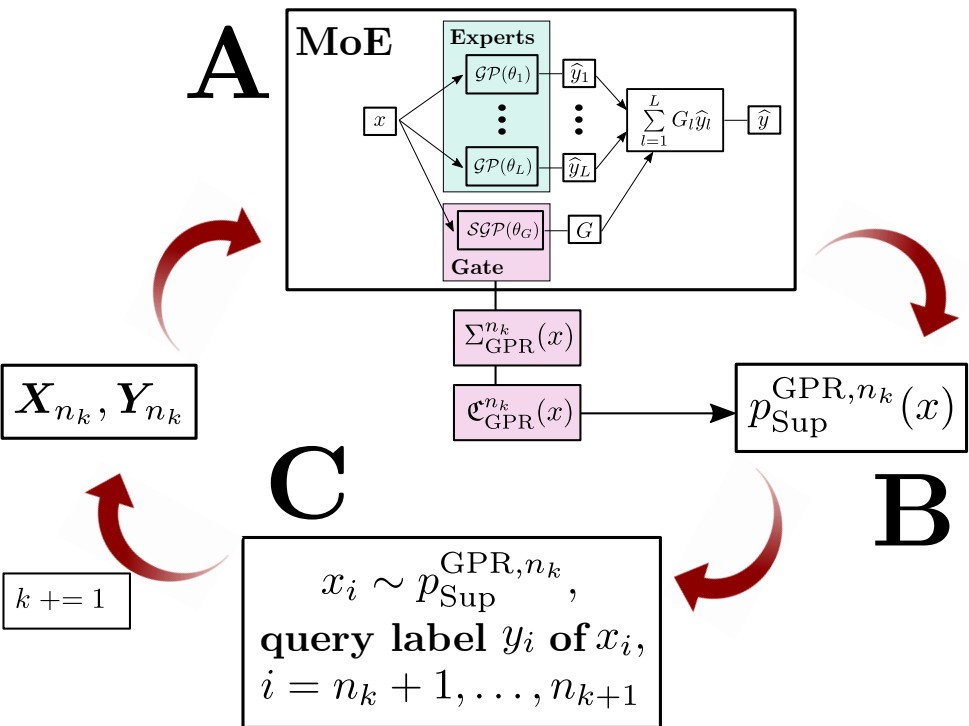

Figure 2: The proposed active learning framework.

### 3.1 Sparse Mixture of Gaussian Processes

Following the work of Shazeer et al. (2017), we construct a sparse MoE model

$$\widehat{f}_{\text{MoE}}(x) = \sum\nolimits_{l=1}^{L} G(x)_l \widehat{y}_l(x), \tag{12}$$

where the gate $G : \mathcal{X} \rightarrow [0,1]^L$ is a probability assignment of an input $x$ to the expert models $\widehat{y}_l$. In particular, it holds $\sum_{l=1}^{L} G(x)_l \equiv 1$ and $G(x)_i \geq 0, \forall x \in \mathcal{X}$ and $1 \leq i \leq L$ (see Appendix B.3 for details).

This MoE approach has two hyperparameters, $\kappa$ and $\mathfrak{s}$, for controlling the sparsity of the gate and adding noise to the gate responses while training in order to escape local optima. We choose the expert models to be (sparse) Gaussian processes (Williams & Rasmussen, 1996; Hensman et al., 2015), that is, $\widehat{y}_l \sim \mathcal{GP}(\theta_{e_l})$ or $\widehat{y}_l \sim \mathcal{SGP}(\theta_{e_l})$, which are parameterized by $\theta_{e_l}$, as described in Appendix B.1 and B.2, and we choose the single channel gating models $g_l \sim \mathcal{SGP}(\theta_{g_l})$ to be sparse Gaussian processes. The overall MoE hyperparameter set is thus given by $\Theta = (\{\theta_{e_l}\}_{l=1}^{L}, \{\theta_{g_l}\}_{l=1}^{L}, \kappa, \mathfrak{s})$.

We will keep certain hyperparameters of $\Theta$ constant after initialization, and share some hyperparameters across experts and the channels of the gate: While the covariances of the inducing value distributions $\boldsymbol{S}^* \in \theta, \theta \in \Theta$ could be full positive definite matrices, we apply $\boldsymbol{S}^* = 0$ throughout, giving favorable stability and computational efficiency.

For the same reasons we choose to fix the inducing point locations $\boldsymbol{X}^* \in \theta, \theta \in \Theta$ after initialization. Furthermore we share the inducing point locations among the experts, respectively the gate channels, such that for $\boldsymbol{X}^* \in \theta_{e_l}$ we apply $\boldsymbol{X}^* = \boldsymbol{X}_E^*$ and for $\boldsymbol{X}^* \in \theta_{g_l}$ we apply $\boldsymbol{X}^* = \boldsymbol{X}_G^*$, for all $1 \leq l \leq L$.

In this work, our goal is to fit a single, coherent regression problem by a MoE approach. Therefore, we propose to share all the parameters across the experts that characterize the regression function rather than the expert model. That is, we share the mean $\mu_E$, the regularization parameter $\lambda_E$ and the noise variance function $\widehat{v}$, respectively the global noise variance $\sigma_\varepsilon^2$ with $\widehat{v}(x) \equiv \sigma_\varepsilon^2$ in case of homoscedasticity. Furthermore, we apply a fixed, logarithmically spaced set of individual expert bandwidth scaling factors $\sigma_1 < \ldots < \sigma_L$

that are multiplied to a fixed, shared bandwidth matrix $\Sigma_E$. Our expert parameters therefore reduce to

$$\theta_{e_l} = (\mu_E, \lambda_E, \widehat{v}, \sigma_l \Sigma_E, \boldsymbol{X}_E^*, \boldsymbol{\mu}_{e_l}^*, 0).$$

For the sake of a simple notation we set $\boldsymbol{X}_E^* = \boldsymbol{\mu}_{e_l}^* = \emptyset$ in the non-sparse GPR case, noting that there are, in fact, no inducing points.

Since our objective does not incorporate any likelihood about the gate's output, there is no noise function to fit for the gate, such that we set $\widehat{v} \equiv 0$ for $\widehat{v} \in \theta_{g_l}$ and all $1 \leq l \leq L$. Each output channel of the gate poses its own classification problem, which is why we do not share the means. Yet, we share the regularization parameter and the bandwidth, as the individual channels should be structurally similar. Our gate parameters therefore reduce to

$$\theta_{g_l} = (\mu_{g_l}, \lambda_G, 0, \sigma_G \mathcal{I}_d, \boldsymbol{X}_G^*, \boldsymbol{\mu}_{g_l}^*, 0).$$

This MoE can cope with a varying structural complexity through the individual bandwidth scaling factors $\sigma_l$ of the experts and heteroscedastic noise through the adaptive regularization. The following subsections will be devoted to the training of our model. We first set up the training objective and describe the training procedure of our model in Sec. 3.1.1, where we identify hyperparameters of the approach. Then, we discuss how to choose the essential hyperparameters systematically on the initial training dataset in Sec. 3.1.2.

### 3.1.1 Training procedure

In this section, we will set up the training objective of our MoE model and discuss which variables of the MoE are parameters to be learned while training, and which variables remain hyperparameters that need to be specified or tuned through an external validation step. We implement our model in *PyTorch* (Paszke et al., 2019), using the *GPyTorch*-package (Gardner et al., 2018). Given the training set $(\boldsymbol{X}_n, \boldsymbol{Y}_n)$, we update the objective described below from in *batch mode* via ADAM-optimization (Kingma & Ba, 2015).

**The main objective** Denote by $\varnothing \subsetneq \mathcal{B} \subseteq \{1, \ldots, n\}$ the indices of a batch. If we assume our problem to be (almost) noise-free, we apply the *mean squared error* (MSE) as the main objective:

$$\text{MSE}(\boldsymbol{X}_n, \boldsymbol{Y}_n, \mathcal{B}, w, \Theta) = w_{\mathcal{B}}^{-1} \sum\nolimits_{b \in \mathcal{B}} w(x_b) \|y_b - \widehat{y}(x_b)\|^2,$$

where $w$ is the training importance weight function and $w_{\mathcal{B}} = \sum\nolimits_{b \in \mathcal{B}} w(x_b)$.

If we assume noise, we instead choose the *negative predictive marginal log-likelihood*

$$\text{MLL}(\boldsymbol{X}_n, \boldsymbol{Y}_n, \mathcal{B}, w, \Theta) = -w_{\mathcal{B}}^{-1} \sum\nolimits_{l=1}^{L} \Big[ \sum\nolimits_{b \in \mathcal{B}} v_l(x_b) P_{b,l} + \tfrac{1}{n_l} \text{KL}(q_l, p_l) \Big]$$

as our main objective, where we have defined

$$n_l = n \sum\nolimits_{b \in \mathcal{B}} w(x_b) \Big/ \sum\nolimits_{b \in \mathcal{B}} v_l(x_b), \quad v_l(x) = G(x)_l w(x),$$

the predictive log-likelihood of the l-th expert in $x_b$ by

$$P_{b,l} = \log \mathbb{E}_{u \sim q_l} \int p_l(y_b|f) p_l(f|u, x_b) df,$$

and $q_l$ is the variational distribution and $p_l$ is the prior distribution of the inducing function values of the expert inducing points $\boldsymbol{X}_E^*$.

**A penalty on small bandwidth choices** In the spirit of Lepski's method (Lepski, 1991; Lepski & Spokoiny, 1997), we prefer the largest choice of bandwidth out of all candidates that perform comparably well. In order to enforce this, we penalize smaller bandwidth choices by adding the following term:

$$\text{pen}_\sigma(\boldsymbol{X}_n, \boldsymbol{Y}_n, \mathcal{B}, w, \Theta) = \frac{2}{(L-1)} \sum\nolimits_{l=1}^{L} \nu_l(L-l) \Big/ \sum\nolimits_{l=1}^{L} \nu_l, \tag{13}$$

where $\nu = \sum_{b \in \mathcal{B}} w(x_b) G(x_b)$. Our total objective then amounts to

$$\text{Obj}(\boldsymbol{X}_n, \boldsymbol{Y}_n, \mathcal{B}, w, \Theta) = \text{MLL}(\boldsymbol{X}_n, \boldsymbol{Y}_n, \mathcal{B}, w, \Theta) + \vartheta_\sigma \text{pen}_\sigma(\boldsymbol{X}_n, \boldsymbol{Y}_n, \mathcal{B}, w, \Theta),$$

or in the noise-free case to

$$\text{Obj}(\boldsymbol{X}_n, \boldsymbol{Y}_n, \mathcal{B}, w, \Theta) = \text{MSE}(\boldsymbol{X}_n, \boldsymbol{Y}_n, \mathcal{B}, w, \Theta) + \vartheta_\sigma \text{pen}_\sigma(\boldsymbol{X}_n, \boldsymbol{Y}_n, \mathcal{B}, w, \Theta).$$

Recall from Sec. 3.1 that the MoE has two further hyperparameters, $\kappa$ for enforcing sparse gate responses and a noise term on the gate responses while training, which is controlled by $\mathfrak{s}$.

Instead of learning $\mathfrak{s}$ as a parameter while training—like proposed by Shazeer et al. (2017)—we propose to shrink $\mathfrak{s} \leftarrow \mathfrak{s}\eta_\mathfrak{s}$ after each training epoch, for a multiplicative factor $\eta_\mathfrak{s} < 1$ and an initial value $\mathfrak{s} := \mathfrak{s}_0$ as hyperparameters. We discuss this heuristic choice in Appendix D.

Finally, note that we require appropriate learning rates for the individual types of tunable parameters: Within a Gaussian process component $\mathcal{SGP}(\theta)$ with $\theta = (\mu, \lambda, \widehat{v}, \Sigma, \boldsymbol{X}^*, \boldsymbol{\mu}^*, \boldsymbol{S}^*)$, the hyperparameters $(\mu, \lambda, \widehat{v})$ must be updated on a smaller scale than the inducing value distribution, given by $(\boldsymbol{\mu}^*, \boldsymbol{S}^*)$. In this regard, let $\eta_H < 1$ be the factor such that, if we update $\boldsymbol{\mu}^*$ at rate $\eta$, then we update $(\mu, \lambda, \widehat{v})$ at rate $\eta_H \eta$.

Similarly, we need to update the gate parameters $\theta_{g_l}$ on smaller scale than the expert parameters $\theta_{e_l}$. In this regard, let $\eta_G < 1$ be the factor such that, if we update $\theta_{e_l}$ at rate $\eta$, then we update $\theta_{g_l}$ at rate $\eta_G \eta$.

The overall set of hyperparameters is thus given by

$$(B, \kappa, \{\sigma_l\}_{l=1}^L, \sigma_G, \lambda_G, \boldsymbol{X}^*, \boldsymbol{X}_G^*, \mathfrak{s}_0, \eta_\mathfrak{s}, \vartheta_\sigma, \eta, \eta_H, \eta_G),$$

whereas the overall set of parameters is given by

$$\Theta = (\mu_E, \lambda_E, \widehat{v}, \Sigma_E, \boldsymbol{\mu}^*, \mu_G, \boldsymbol{\mu}_G^*).$$

We provide further details on the design choices for our MoE model in Appendix D.

### 3.1.2 Choosing the hyperparameters

While the set of hyperparameters appears to be large, most of them can be tuned in advance on the initial training dataset of moderate size and can be held fixed in the subsequent training data refinement process. In addition, some hyperparameters $(|\boldsymbol{X}^*|, |\boldsymbol{X}_G^*|, \kappa)$ are not crucial for our approach to work, as long as they are not underestimated, where smaller values are desirable with regard to computational cost.

- The MoE is robust with respect to unnecessarily large choices of the gate output sparsity $\kappa$, and so, we can initialize $\kappa = L$ while choosing the remaining hyperparameters, followed by tuning $\kappa$ as the last hyperparameter, where we successively reduce $\kappa$ until we observe a significant loss of performance of the MoE.

- The numbers $m_E = |\boldsymbol{X}^*|$, $m_G = |\boldsymbol{X}_G^*|$ of inducing points of the expert and the gate models are the main driver of the computational complexity of our MoE. While unnecessarily large numbers will not hurt the model performance, they should therefore be set to the smallest value that leads to no significant loss of performance to keep the computational complexity of the model moderate at larger training sizes. We suggest to use $m_E = n_0$ for the experts and $m_G = \frac{n_0}{4}$ in the initial iteration. After having a first estimate of LFC and a refinement of the initial training data, their number can be adapted via validation of the MoE in the second iteration, and held fixed afterwards.

- Note that the choice of the locations of the inducing points $\boldsymbol{X}^*, \boldsymbol{X}_G^*$ are discussed in Appendix C, which are not subject to optimization in our work.

Let us begin with the base learning rate and the expert's internal hyperparameters learning rate $(\eta, \eta_H)$ and the batch size $B$ that are related to a single GPR expert rather than the whole MoE model. Generally, we suggest to apply an adaptive base learning rate, where we shrink $\eta_i \leftarrow \frac{1}{2}\eta_{i-1}$ with $\eta_0 = \eta$ while training as soon as the validation performance gets stuck, until $\eta_i$ crosses a lower threshold like, e.g., $\eta_i < \eta_0/1000$.

- First, we hold $\eta_H = 0.2$, $B = n_0$ fixed at reasonable initial values to train and validate a single, isotropic, sparse GPR expert with respect to $\eta$. Here, too small values of $\eta$ lead to a very slow convergence of the objective, in which case we interrupt the training immediately. We then increase $\eta$ as long as the first objective updates are consistently decreasing.

- Next, we validate $\eta_H$, where we gradually decrease from $\eta_H = 1$. Again, we interrupt the training for too large choices of $\eta_H$, where the training objective will diverge.

- Finally, we validate $B$, where we gradually decrease from $B = n_0$.

Next, we have to deal with the remaining hyperparameters that are related to the MoE model. For this, we first initialize the less crucial hyperparameters at reasonable values, tuning them afterwards: We apply a set of experts with $\{\sigma_l\}_{l=1}^L$, where $L = 7$, and $\sigma_l = \widehat{\sigma}_{\mathrm{GPR}} 2^{\frac{l-4}{d}}$, which are logarithmically spaced around the global bandwidth estimate $\widehat{\sigma}_{\mathrm{GPR}}$ of the best performing model that we obtained from the above tuning of the hyperparameters related to a single GPR expert. The noise added to the gate $(\mathfrak{s}_0, \eta_\mathfrak{s})$ as well as the regularization $\vartheta_\sigma$ of the bandwidth function are about fine-tuning of the model. We set them to $\mathfrak{s}_0 = 0.1, \eta_\mathfrak{s} = 1/\sqrt{2}$ and $\vartheta_\sigma = 0.01$. Note that we suggest to keep $\eta_\mathfrak{s} = 1/\sqrt{2}$ throughout without further tuning.

- We validate $\sigma_G, \lambda_G$ over a grid, where we gradually decrease the gate learning rate from $\eta_G = 1$.

As a last step, we choose the hyperparameters for fine-tuning of the MoE model:

- First, we validate the MoE for $\vartheta_\sigma$

- Next, we tune $\{\sigma_l\}_{l=1}^L$: First, we observe that unreasonable bandwidths will be automatically dropped while training. Therefore, if the minimal or maximal candidate associated to $\sigma_1, \sigma_L$ is not chosen while training, we remove the respective expert and retrain the MoE. Vice versa, we expand the bandwidth candidate range beyond $\sigma_1, \sigma_L$ with a factor of $2^{\mp\frac{1}{d}}$ as long as the boundary candidates are not dropped while training.

- Finally, we validate the MoE for $\mathfrak{s}_0$

## 3.2 GPR-based estimates of LOB and LFC

Inspired by the generalized results to LFC and the *superior training density* of LPS in Eq. (9) and (10), we are now able to deduce their GPR-based analog: This result deals with models in combination with a bandwidth space $\mathcal{S}$ that adapt universally[2] to functions $f \in \mathcal{C}^\alpha(\mathcal{X}, \mathbb{R})$—which is the case for GPR. The idea of the LFC estimate was to adjust LOB appropriately so that it becomes invariant with respect to the influence of the training density, heteroscedasticity and its global decay with respect to the training size $n$. After training of the model from Sec. 3.1, we use the learned gate $G$ from (12) to predict LOB of GPR as

$$\Sigma_{\mathrm{GPR}}^n(x) = \sigma_{\mathrm{GPR}}^n(x)\Sigma_E, \quad \text{where} \quad \sigma_{\mathrm{GPR}}^n(x) = \exp\left\{\sum_{l=1}^L G(x)_l \log(\sigma_l)\right\}. \tag{14}$$

Combining the local effective sample size $p(x)n$ with the scaling result of $\Sigma_{\mathrm{GPR}}^n$ in (11) from Sec. 2.3, we propose an LFC estimate for GPR as follows:

---

[2]That is, the MISE decays at the minimax-rate $n^{-\frac{2\alpha}{2\alpha+d}}$ of nonparametric models.

**Definition 1** (LFC of GPR). *For $f \in \mathcal{C}^\alpha(\mathcal{X}, \mathbb{R})$, $\boldsymbol{X}_n \sim p$ and homoscedastic noise, our GPR-based LFC estimate of $f$ in $x \in \mathcal{X}$ is asymptotically given by*

$$\mathfrak{C}_{GPR}^n(x) = \left[\frac{1}{p(x)n}\right]^{\frac{d}{2\alpha+d}} |\Sigma_{GPR}^n(x)|^{-1} = \left[\frac{1}{p(x)n}\right]^{\frac{d}{2\alpha+d}} \sigma_{GPR}^n(x)^{-d}. \tag{15}$$

In analogy to Eq. (9) $\mathfrak{C}_{\mathrm{GPR}}^n$ measures the structural complexity of $f$, as it asymptotically does not depend on $p$, $v$ and $n$. Note that from the theoretical side we have made the restriction of homoscedastic noise in the definition of $\mathfrak{C}_{\mathrm{GPR}}^n$, since we are not aware of a theory on the scaling of GPR-based LOB with respect to heteroscedasticity. Nevertheless, as opposed to GPR, the LPS model provides no explicit way to adapt to the local noise variance $v(x)$, such that the LOB of LPS scales with respect to $v$ to address heteroscedasticity. In contrast, when deploying a heteroscedastic GPR model that adapts for $v$ via the regularization, we observe only very little influence of heteroscedasticity on LOB function, which we will demonstrate in Sec. 5.1.

Now, when putting $\mathfrak{C}_{\mathrm{GPR}}^n$ into Eq. (10), we obtain the *superior training density*

$$p_{\mathrm{Sup}}^{\mathrm{GPR},n}(x) \propto [\mathfrak{C}_{\mathrm{GPR}}^n(x)q(x)]^{\frac{2\alpha+d}{4\alpha+d}} v(x)^{\frac{2\alpha}{4\alpha+d}}(1+o(1)). \tag{16}$$

While $p_{\mathrm{Sup}}^{\mathrm{GPR},n}$ will not be optimal for our model, we expect it to be asymptotically superior to the naive *random test sampling*, i.e., $\boldsymbol{X}_n \sim q$, due to the model-agnosticity of the LPS-based result. In order to assess the asymptotic superiority of a training density $p$ (such as $p_{\mathrm{Sup}}^{\mathrm{GPR},n}$) quantitatively, we define the following:

**Definition 2.** *Over the space of square-integrable functions $f \in \mathcal{L}^2(\mathcal{X})$, for a nonparametric regression model $\widehat{f}$ and a training density $p$, we define by $\varrho(\widehat{f}, p) > 0$ the relative required sample size such that for $n' = \varrho(\widehat{f}, p)n$, $\boldsymbol{X}_{n'}' \sim p$ and $\boldsymbol{X}_n \sim q$ it holds that*

$$MISE\left(q, \widehat{f}|\boldsymbol{X}_n\right) = MISE\left(q, \widehat{f}|\boldsymbol{X}_{n'}'\right)(1+o(1)).$$

For a nonparametric model defined over $\mathcal{L}^2(\mathcal{X})$, the law of the MISE does not change with respect to $p$, except for a constant multiple. Now, if $\varrho(\widehat{f}, p) < 1$ it means that we achieve the same performance as *random test sampling* with only a fraction of the number of training samples. In Sec. 5 we will demonstrate the superiority of the training density $p_{\mathrm{Sup}}^{\mathrm{GPR},n}$ for our GPR-based MoE model.

Note that, if $\alpha \to \infty$, the estimates in Eq. (15) and (16) converge to the simpler form

$$\mathfrak{C}_{\mathrm{GPR}}^n(x) = \sigma_{\mathrm{GPR}}^n(x)^{-d}, \quad \text{and} \quad p_{\mathrm{Sup}}^{\mathrm{GPR},n}(x) \propto [\mathfrak{C}_{\mathrm{GPR}}^n(x)q(x)v(x)]^{\frac{1}{2}}. \tag{17}$$

Besides being an ingredient to active learning, LFC can also be used to reduce the required model complexity. For example, in an RBF-network or a sparse GPR model, we can coarsen or refine the model resolution by placing an adequate amount of basis functions or inducing points, respecting LFC. We will discuss this choice in Appendix C and demonstrate its ability to reduce the overall model complexity in Sec. 5.1. Finally, LFC can be inspected to obtain deeper insights into the research field of the regression problem, which is particularly hard for high-dimensional data (see Sec. 5.2).

### 3.3 The active learning framework

Starting with an initial training set $\boldsymbol{X}_{n_0}, \boldsymbol{Y}_{n_0}$ of size $n_0$ with $\boldsymbol{X}_{n_0} \sim p_0$ for some initial training distribution such as $p_0 \equiv q$, we implement the online sampling procedure as described in Panknin et al. (2021), such that $\boldsymbol{X}_n \sim p_{\mathrm{Sup}}^{\mathrm{GPR},n}$ as $n \to \infty$. We grow the training set as follows:

Given the current training set $\boldsymbol{X}_{n_k}, \boldsymbol{Y}_{n_k}$ we estimate $\widehat{\Sigma}_{\mathrm{GPR}}^{n_k}$ as described in Sec. 3.2. Using (15), (16), it is

$$\widehat{\mathfrak{C}}_{\mathrm{GPR}}^{n_k}(x) \propto [1/p_k(x)]^{\frac{1}{2\alpha+d}} \left|\widehat{\Sigma}_{\mathrm{GPR}}^{n_k}(x)\right|^{-1}, \quad \text{and}$$

$$\widehat{p}_{\mathrm{Sup}}^{\mathrm{GPR},n_k}(x) \propto \left[\widehat{\mathfrak{C}}_{\mathrm{GPR}}^{n_k}(x)q(x)\right]^{\frac{2\alpha+d}{4\alpha+d}} v(x)^{\frac{2\alpha}{4\alpha+d}}.$$

Note that, letting the next sample size be $n_{k+1} = 2n_k$, we have already drawn half the samples of $n_{k+1}$ according to a potentially different distribution $p_k$ than the new proposed $\widehat{p}_{\mathrm{Sup}}^{\mathrm{GPR}, n_k}$. The closest we can get in distribution to $\widehat{p}_{\mathrm{Sup}}^{\mathrm{GPR}, n_k}$ is given by $\boldsymbol{X}_{n_{k+1}} \sim p_{k+1}$, where $p_{k+1} := \gamma_2 p_k + (1 - \gamma_2)\widehat{p}_{\mathrm{Sup}}^{\mathrm{GPR}, n_k}$, for $\gamma_2 = \max\left\{0, \dfrac{0.5 - \gamma_1^{-1}}{1 - \gamma_1^{-1}}\right\} \in [0, 0.5)$ and $\gamma_1 = \max\limits_{x \in \mathcal{X}} \dfrac{p_k(x)}{\widehat{p}_{\mathrm{Sup}}^{\mathrm{GPR}, n_k}(x)}$. This is achieved by sampling $x_{n_k+1}, \ldots, x_{n_{k+1}} \sim \widetilde{p}_{k+1}$ for $\widetilde{p}_{k+1} = 2p_{k+1} - p_k$, which is a valid probability density (Panknin et al., 2021).

**Adaptions in the pool-based active learning scenario**  In the active learning framework described above, we deal with properly normalized probability densities. But in the *pool-based* active learning scenario such normalization is usually impossible, since our information about the input space $\mathcal{X}$ is restricted to a large, unlabeled *pool* of samples $X_{\mathrm{MD}} \in \mathcal{X}^N$. This pool follows a distribution $X_{\mathrm{MD}} \sim p_{\mathrm{MD}}$, for which it is common to assume an (unnormalized) density estimate $\widehat{p}_{\mathrm{MD}}$ to be given: Unlabeled inputs are considered cheaply accessible, whereas querying labels is expensive.

Now, for our framework to be applicable, it suffices to keep all considered densities such as $\widehat{p}_{\mathrm{Sup}}^{\mathrm{GPR}, n_k}$ at equal norm, which we can enforce via normalizing a density $p$ by $\widetilde{p} = p/\mathrm{norm}(p)$, where

$$\mathrm{norm}(p) = |X_{\mathrm{MD}}|^{-1} \sum\nolimits_{x \in X_{\mathrm{MD}}} p(x)/\widehat{p}_{\mathrm{MD}}(x).$$

To see this, note that first of all $\widehat{p}_{\mathrm{MD}}$ is an unnormalized estimate to $p_{\mathrm{MD}}$ such that we can write $\widehat{p}_{\mathrm{MD}} \approx c \cdot p_{\mathrm{MD}}$ for some unknown constant $c > 0$. On the one hand it is $\int_{\mathcal{X}} \widehat{p}_{\mathrm{MD}}(x)dx = c$ by definition. On the other hand it is

$$\mathrm{norm}(p) \approx \int_{\mathcal{X}} \frac{p(x)}{\widehat{p}_{\mathrm{MD}}(x)} p_{\mathrm{MD}}(x)dx = \frac{1}{c} \int_{\mathcal{X}} p(x)dx,$$

such that also

$$\int_{\mathcal{X}} \widetilde{p}(x)dx = \frac{1}{\mathrm{norm}(p)} \int_{\mathcal{X}} p(x)dx \approx c$$

holds for any unnormalized density $p$.

Subsequently, the required samples $x_{n_k+1}, \ldots, x_{n_{k+1}} \sim \widetilde{p}_{k+1}$ are obtained via *importance sampling* from the pool with *importance weights* $\widetilde{p}_{k+1}(x)/[\mathrm{norm}(\widetilde{p}_{k+1})\widehat{p}_{\mathrm{MD}}(x)]$ for $x \in X_{\mathrm{MD}}$.

## 4  Discussion of model construction and related work

First of all, let us recall that we have chosen to share all parameters across the experts that describe a single regression function. In contrast, the common assumption of MoE approaches is that the overall problem to infer is too complex for a single, comparably simple expert. This is the case, for example in regression of nonstationary or piecewise continuous data, and naturally in classification where each cluster shape may follow its own pattern. In such a scenario each expert of the MoE model can specialize on modelling an individual, (through the lens of a single expert) incompatible subset of the data, where the gate learns a soft assignment of data to the experts. Under these assumptions the hyperparameters of each expert can be tuned individually on the respective assigned data subset. In the light of this paradigm there exist several instances of mixture of Gaussian processes, for example Tresp (2001); Meeds & Osindero (2006); Yuan & Neubauer (2009); Yang & Ma (2011); Chen et al. (2014).

Now that we assume a single regression problem, there is no such segmentation as described above: Each individual (reasonably specified) expert of our mixture model is eventually capable of modelling the whole problem on its own. Yet, if the problem possesses an inhomogeneous structure, the prediction performance can be increased by allowing for a locally individual bandwidth choice. This less common assumption was also made by Pawelzik et al. (1996), where—locally dependent—some experts are expected to perform superior compared to the others.

There exist other sparse GPR approaches that could be considered for the gate or the experts of our MoE model. Some are computationally appealing as they solve for the inducing value distribution analytically

(Seeger et al., 2003; Titsias, 2009) as opposed to the variational, stochastic gradient descent based method we deployed (Hensman et al., 2015). For the gate however, this kind of approach comes with complications as it requires labels. Such labels do not exist for the gate, and so we would need so train the MoE in an *expectation maximization* loop, where the likelihood of an expert to have produced a training label functions as a pseudo-label to the gate. For the experts on the other hand, any choice of GPR model is possible.

Recall from Appendix C that we choose the inducing point locations of the gate and the experts of our MoE model in a diverse and representative way, but also in alignment with the structural complexity of the target function, interpreting this choice as a nested AL problem. Of course, there are a variety of inducing point selection approaches in the literature. Zhang et al. (2008) interpreted the choice of inducing point locations from a geometric view that is similar to ours: They derived a bound on the reconstruction error of a full kernel matrix by a Nyström low-rank approximation in terms of the sum of distances of all training points to their nearest inducing point. This exposes a local minimum by letting the inducing point locations be the result of *k-Means clustering*. This choice of inducing point locations is representative and diverse, while it solely considers input space information. In this sense, our approach extends their work by additionally considering label information. This and our approach draw a fixed number of inducing points at once. There are also a lot of Nyström method based inducing point selection approaches that select columns of the full kernel matrix according to a fixed distribution (Drineas et al., 2005) or one-by-one in a greedy, adaptive way (Smola & Schölkopf, 2000; Fine & Scheinberg, 2001; Seeger et al., 2003). An intensive overview of Nyström method based inducing point selection methods was given by Kumar et al. (2012), where they also analyzed ensembles of low-rank approximations.

In this work, we consider model-agnostic active learning with persistent performance at large (or even asymptotic) training size. In this sense, we delimit us from active learning approaches that are tied to a model, e.g., when they are based on a parametric model Kiefer (1959); MacKay (1992); He (2010); Sugiyama & Nakajima (2009); Gubaev et al. (2018), or which build up training data in a greedy way to maximize its information content at small sample size, where the information is either based on the inputs only (Teytaud et al., 2007; Yu & Kim, 2010; Wu, 2019; Liu et al., 2021) or also incorporates the labels Seo et al. (2000); Burbidge et al. (2007); Cai et al. (2013). Note that depending on the use-case, such approaches can exceed the performance of our approach. For example, if there is enough domain knowledge such that we can deduce a reasonable parametric model without the need of a model change in hindsight, an active sampling scheme based on this model will be best. Our category of interest is for the other case, when domain knowledge is scarce, where we have no idea about the regularity or structure of the problem to decide on a terminal model. Panknin et al. (2021) already have compared favorably to related work within this category. They compared to Goetz et al. (2018) in a heteroscedastic setting, using a regression forest model and to Bull et al. (2013) in a setting of inhomogeneous complexity, using an RBF-network. This demonstrates the flexibility of this AL approach in terms of learning problem specifications as well as model choices. As we extend on their work, we claim this favorable performance by transitivity.

There exists a lot of research on active learning for GPR (Seo et al., 2000; Pasolli & Melgani, 2011; Schreiter et al., 2015; Yue et al., 2020), which is typically based on minimizing prediction uncertainties of the model. Now, with our proposed active learning approach being based on a mixture of GPR models, this research area is the most related competitor to our work.

For a standard GPR model, the prediction uncertainty is the higher the farther away we move from training inputs. In this way, *GP uncertainty sampling* does not depend on the regression function to infer and is instead an input space geometric argument: Uncertainty sampling makes up for a *low-dispersion sequence* (Niederreiter, 1988). In this sense, it samples (pseudo-)uniformly from the input space, which is inferior to following the distribution of our proposed sampling scheme, as we will show in Sec. 5.1. Using the fact that our approach is a mixture of GPR models, we can come up with a straight-forward extension of GP uncertainty sampling which takes the inhomogeneous complexity of data into account: We can derive a *mixture of Gaussian process uncertainties* (MoGPU) by simply weighting the predictive variances of all experts in some input $x$ with respect to the gate values $G(x)$ from (12) as

$$\text{MoGPU}(x) = \sum_{l=1}^{L} G(x)_l \boldsymbol{C}_{\theta_l}^*(x),  \tag{18}$$

where $C_{\theta_l}^*$ is the predictive variance of the l-th expert (see (28)). We consider MoGPU as a baseline competitor to our approach, which is itself superior to GP uncertainty sampling as it can cope with structural inhomogeneities. We will show that our approach outperforms MoGPU (and therefore implicitly GP uncertainty sampling) and discuss the difference between MoGPU and our approach on synthetic data in Sec. 5.1.

## 5 Experiments

In this section, we will first analyze our approach on toy-data, regarding the MoE model, LFC and the *superior training density*. Here, we also compare to related work. Then, we apply our approach to a high-dimensional MD simulation dataset from quantum chemistry, by which we can deduce deeper insights into this regression problem.

First, let us denote the *mean absolute error* (MAE), the *maximum absolute error* (max AE) and the *root mean squared error* (RMSE) of a model $\widehat{f}$ for a test set $\boldsymbol{X}_N^{\mathrm{T}} \subset \mathcal{X}^N$ with $\boldsymbol{X}_N^{\mathrm{T}} \sim q$ by

$$\mathrm{MAE}(\widehat{f}, \boldsymbol{X}_n, \boldsymbol{Y}_n) = N^{-1} \sum\nolimits_{x \in \boldsymbol{X}_N^{\mathrm{T}}} \left| f(x) - \widehat{f}(x) \right|, \ \ \max \mathrm{AE}(\widehat{f}, \boldsymbol{X}_n, \boldsymbol{Y}_n) = \max_{x \in \boldsymbol{X}_N^{\mathrm{T}}} \left| f(x) - \widehat{f}(x) \right| \ \ \text{and}$$

$$\mathrm{RMSE}(\widehat{f}, \boldsymbol{X}_n, \boldsymbol{Y}_n) = \left[ N^{-1} \sum\nolimits_{x \in \boldsymbol{X}_N^{\mathrm{T}}} \left| f(x) - \widehat{f}(x) \right|^2 \right]^{\frac{1}{2}} .$$

For our model $\widehat{f}_{\mathrm{MoE}}$ and $f \in \mathcal{C}^\alpha(\mathcal{X}, \mathbb{R})$, we can estimate the relative sample size that is required for samples drawn according to $p$ to perform on par with samples drawn according to $q$, as defined in Definition 2 , by

$$\varrho(\widehat{f}_{\mathrm{MoE}}, p) \approx \left[ \frac{\mathrm{RMSE}(\widehat{f}_{\mathrm{MoE}}, \boldsymbol{X}_n', \boldsymbol{Y}_n')}{\mathrm{RMSE}(\widehat{f}_{\mathrm{MoE}}, \boldsymbol{X}_n, \boldsymbol{Y}_n)} \right]^{2\beta} \ , \text{with} \quad \beta = \begin{cases} \frac{2\alpha+d}{2\alpha}, & \alpha < \infty \\ 1, & \alpha = \infty \end{cases}, \tag{19}$$

where it is $\boldsymbol{X}_n' \sim p$ and $\boldsymbol{X}_n \sim q$ with respective labels $\boldsymbol{Y}_n'$ and $\boldsymbol{Y}_n$. Setting $p = \widehat{p}_{\mathrm{Sup}}^{\mathrm{GPR},n}$, we can quantify the active learning performance of our proposed active learning framework in the following experiments.

### 5.1 Doppler function

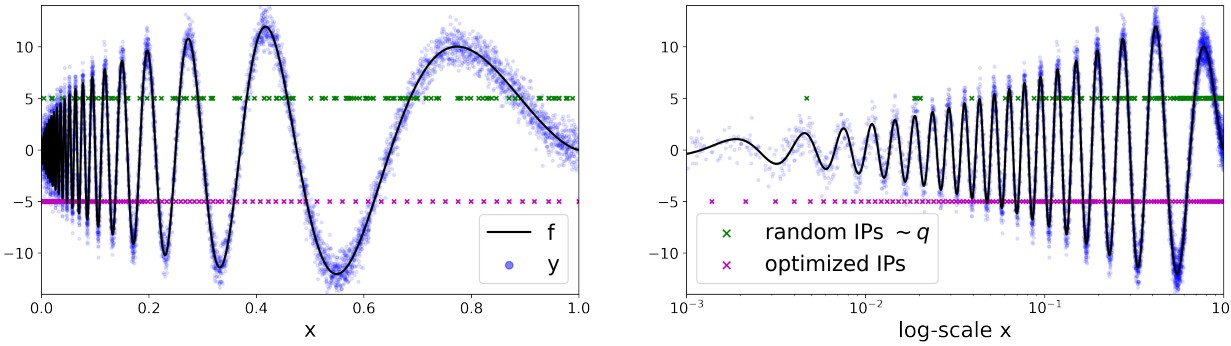

Figure 3: The Doppler experiment: An exemplary dataset and the locations of 128 inducing points, once sampled most naively—that is, random according to the test distribution—and once optimized regarding diversity as well as structural complexity and representativeness, shown on natural x-scale (left) and on logarithmic x-scale (right).

We will first demonstrate our approach on the *Doppler* function (see, for example, Donoho & Johnstone (1994)), which was also discussed in related work that deals with inhomogeneous complexity Panknin et al. (2021); Bull et al. (2013). This one-dimensional, homoscedastic toy-example allows for an easy and intuitive visualization. For $x \in \mathcal{X} = [0,1]$, let

$$\mathbb{P}(y|x) = \mathcal{N}(y; f(x), 1), \quad f(x) = C\sqrt{x(1-x)} \sin\left(2\pi(1+\epsilon)/(x+\epsilon)\right),$$

where $\epsilon = 0.05$, $C$ is chosen such that $\|f\|_2 = 7$ and $\mathcal{N}(\cdot; \mu, \sigma^2)$ denotes the Gaussian distribution with mean $\mu$ and variance $\sigma^2$. Fig. 3 shows an example dataset as blue dots and the true function $f$ to infer in black. We assume a uniform test distribution, such that $q \sim \mathcal{U}(\mathcal{X})$.

Due to the strong variation of structural complexity, a single-scale GPR model does not cope well with the Doppler function, as we discuss in Appendix E.1.

We implement our proposed MoE model as described in Sec. 3.1 with sparse Gaussian processes as the expert and gate models and using the Gaussian kernel $k$. We apply 512, respectively 128 inducing points for the experts and the gate, which are chosen via SVGD (see Appendix C). Furthermore we apply $\sigma_j = 10^{(j-10)/3}, 1 \leq j \leq 7$, as the expert bandwidths, $\lambda_E = 20$ as the initial expert regularization, and $\sigma_G = 0.05$ and $\lambda_G = 10$ for the gate. For the training, we apply a batch size of $B = 512$, a terminal expert sparsity $\kappa = 2$, a penalty factor of $\vartheta_\sigma = 0.5$ for small bandwidth choices, gate noise parameters $\mathfrak{s}_0 = 0.1$ and $\eta_{\mathfrak{s}} = 1/\sqrt{2}$, and learning rate parameters $\eta = 0.01$, $\eta_H = 0.2$, $\eta_G = 1$.

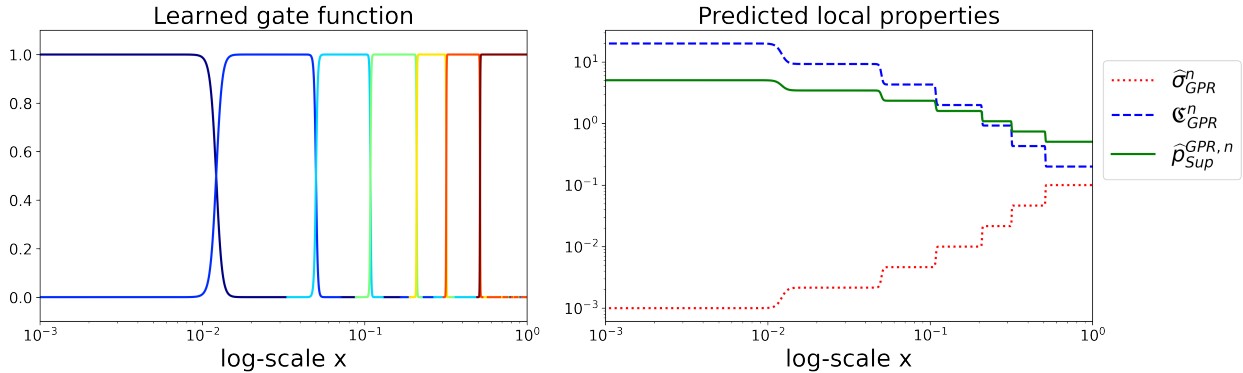

Figure 4: The Doppler experiment: The gate function (left) and the associated estimates to LOB, LFC and *superior training density* (right) trained on the dataset from Fig. 3 and shown on a logarithmic x-scale.

Fig. 4 shows the gate function after training of the MoE model, as described in Sec. 3.1.1, and the associated estimates to LOB, LFC and the *superior training density*, calculated according to (14) and (17).

**Necessity of the small bandwidth penalty**  We impose a penalty on small bandwidth choices through the factor $\vartheta_\sigma = 0.5$ to regularize the bandwidth function and prevent overfitting, as described in Appendix D. We demonstrate this overfitting issue in Appendix E.2 that results from applying no regularization ($\vartheta_\sigma = 0$).

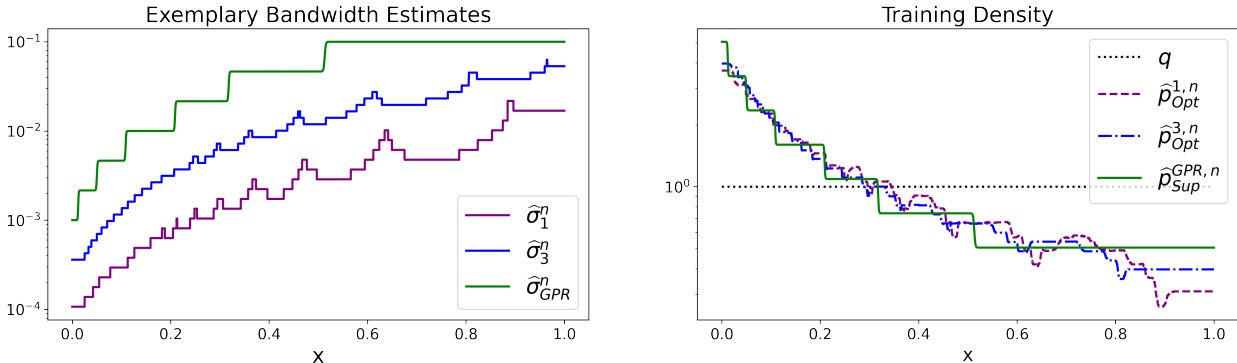

Figure 5: The Doppler experiment: The LOB estimates (left) and the resulting *superior training density* of our proposed GPR-based approach in comparison to the LPS-based approach of order $Q = 1, 3$ (right). The results are averaged over 20 repetitions.

**Parsimonious modelling using LFC**  In Sec. 3.2 we mentioned that LFC can also be used to coarsen or refine the model resolution adequately to reduce the overall complexity of the model. While we fixed the inducing points to reasonable numbers in the other parts of the Doppler experiment, that is, $m = 512$ and

$m = 1024$ inducing points under active, respectively *random test sampling*, we here investigate the influence of the number of inducing points and their distribution on the capability to resemble the Doppler function. Recall from Appendix C that we interpret the choice of the inducing points as a nested AL task at small sample size ($m \ll n$), where it is reasonable for them to be samples in a diverse way, respecting the training distribution but also the structural complexity of the target function. In Fig. 3 (right) we show a naive choice and our optimized choice of inducing points.

In Fig. 6 we compare the RMSE for the fixed training size $n = 2^{15}$ for both settings, active and passive, when sampling the inducing points according to the training density $p$, the LFC and their geometric mean $\sqrt{p \cdot \widehat{\mathfrak{C}}^n_{\mathrm{GPR}}}$. First of all, we observe that we generally require less inducing points with active sampling compared to *random test sampling*, which originates from the fact that the *superior training density* $\widehat{p}^{\mathrm{GPR},n}_{\mathrm{Sup}}$ already respects LFC to some degree. Next, we observe that the geometric mean of the training density and LFC performs best, provided that the number of inducing points $m$ is large enough. Finally, we observe that, non surprisingly, we are able to shrink $m$ the most under the LFC distribution, namely to $m = 128$, before the performance of the model degrades substantially.

In summary, we are able to shrink the model complexity up to a factor of 8 for the Doppler function without a significant loss of performance, when respecting LFC in the model design.

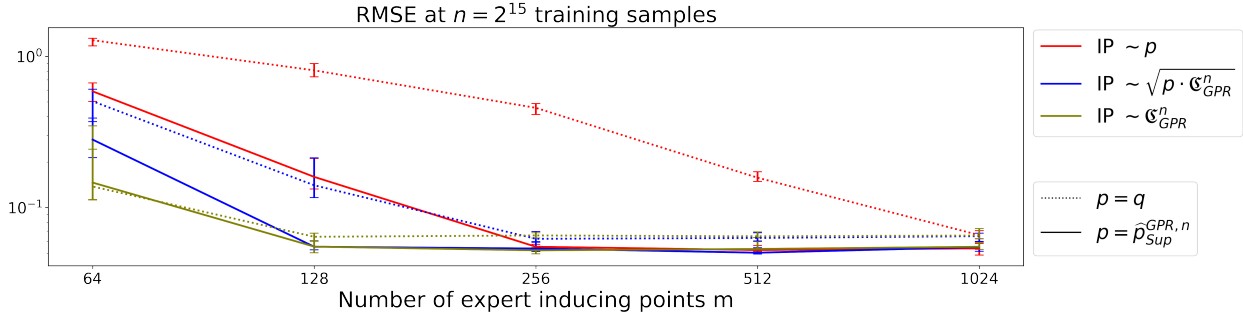

Figure 6: The Doppler experiment: The curves show the RMSE at training size $n = 2^{15}$ for a varying number of expert inducing points $m$. The colors correspond to different inducing point distributions, whereas the line styles correspond to the underlying training distribution. The results are averaged over 20 repetitions.

**Comparing the active learning framework in the LPS and GPR domain** Since $f \in \mathcal{C}^\infty(\mathcal{X}, \mathbb{R})$, our deduced *superior training density* estimate is given by Eq. (17). In Fig. 5 we plot our estimates of LOB and the *superior training density* in comparison to the LPS-based results for polynomial degrees of order $Q = 1, 3$, and with implementation and hyperparameters as described in Panknin et al. (2021). Here, we can observe the qualitative similarity of the LPS- and GPR-based estimates of LOB.

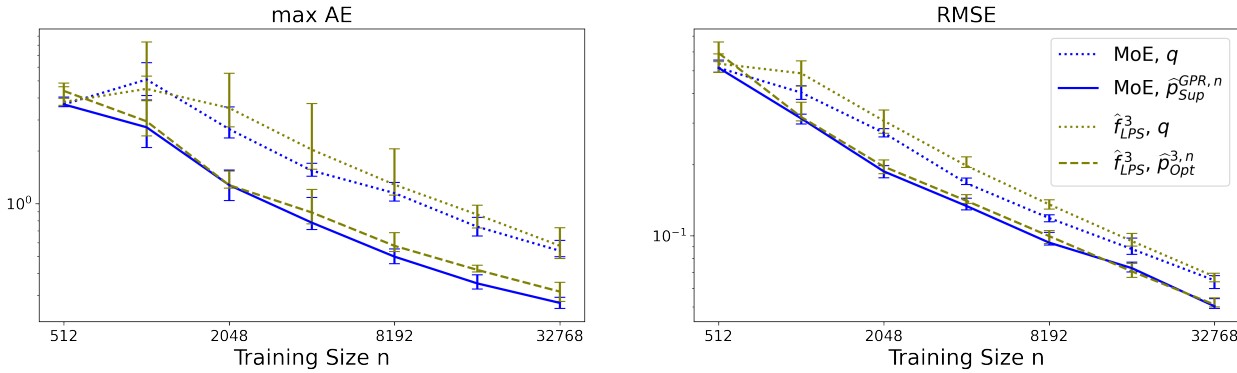

Figure 7: The Doppler experiment: The max AE (left) and the RMSE (right) of our proposed GPR-based approach in comparison to the LPS-based approach of order $Q = 3$ (see Eq. (5) and (6)), once using the respective AL scheme and once, applying *random test sampling*. The results are averaged over 20 repetitions.

When conducting the proposed GPR-based active sampling scheme as described in Sec. 3.2, we additionally observe quantitative benefits in Fig. 7 over *random test sampling*—quite similar to the LPS-based result for $Q = 3$: When estimating the relative sample size (19) we require to achieve the same RMSE via active sampling compared to *random test sampling*, we obtain $\varrho(\widehat{f}_{\mathrm{MoE}}, \widehat{p}_{\mathrm{Sup}}^{\mathrm{GPR},n}) = 0.64 \pm 0.04$. This means that we save more than one third of samples via our active sampling scheme.

This provides evidence for the effectiveness of our proposed active learning framework, combining the theoretical foundation of the LPS domain with the efficient access to LOB estimates in the GPR domain.

**Heteroscedastic noise treatment**   While the treatment of heteroscedastic noise is not the focus of this work, we will now demonstrate our approach on a heteroscedastic version of the Doppler experiment. For this, we let $v(x) = (3 - 4\,|x - 0.5|))^2 \in [1, 9]$, which we plot in Fig. 8 (top left) together with the resulting dataset. Here, we assume the local noise variance (or an estimate to it) to be provided externally, again,

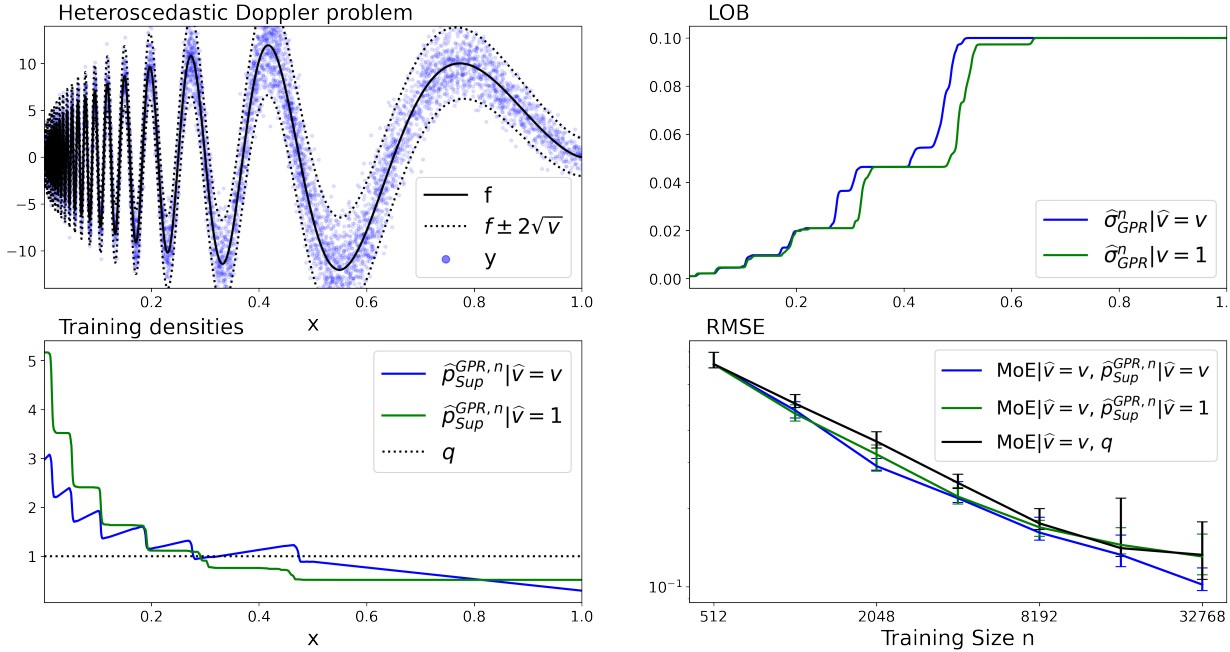

Figure 8: The Doppler experiment under heteroscedastic noise: (Top left) An exemplary dataset; (Top right) The LOB estimates, when comparing the homoscedastic to the heteroscedastic Doppler experiment; (Bottom left) The training densities of *random test sampling*, $p_{\mathrm{Sup}}^{\mathrm{GPR},n}$ and $p_{\mathrm{Sup}}^{\mathrm{GPR},n}$ when wrongly assuming homoscedasticity; (Bottom right) The RMSE at several training sizes of the compared sampling schemes. The results are averaged over 20 repetitions.

since its estimation is out of the scope of this work. However, note that the estimation of $v$ is well-studied in the literature, especially for Gaussian processes Kersting et al. (2007); Cawley et al. (2006); Tresp (2001).

In Fig. 8 (top right) we compare the LOB estimates obtained from the homoscedastic dataset and the heteroscedastic version. As we have suggested in Sec. 3.2, the influence of $v$ on the LOB estimates obtained from heteroscedastic GPR experts is relatively small. Likewise, we proceed with the evaluation of our proposed AL scheme under heteroscedasticity. Here, we compare to *random test sampling*, but also to $p_{\mathrm{Sup}}^{\mathrm{GPR},n}$ under the wrong assumption of homoscedastic noise. Note that we use the heteroscedastic MoE in all cases, since the wrong assumption of homoscedastic noise in the experts makes the MoE very volatile. The respective training densities and RMSE learning curves can be seen in the bottom row in Fig. 8. Due to the stronger noise (compared to the homoscedastic experiment), the asymptotic behavior begins to materialize later from $n = 2^{13}$ training samples. Until this point, sampling only with respect to the structural complexity looks also promising. However, as soon as the target function is roughly resembled, respecting the inhomogeneity in the noise level becomes crucial to achieve a homogeneous pointwise convergence and, thus, maintain asymptotic superiority.

**On Gaussian process uncertainty**  In Sec. 4 we mentioned that GP uncertainty sampling is the most related approach to our proposed framework, since both build on GPR models. However, standard GP uncertainty sampling generates uniformly distributed training data with advantages at small sample sizes, where the chosen samples are spread out more evenly than for actual uniform random sampling.

**Remark 3.** *The training inputs $\boldsymbol{X}_n$ sampled according to GP uncertainty sampling possess the* low-dispersion property*: The dispersion, given by $\sup_{x \in \mathcal{X}} \min_{1 \leq i \leq n} \|x - x_i\|$ (Niederreiter, 1988) is a measure of how well spread out the training sample is. We say that a sequence has low-dispersion if its dispersion is lower that the dispersion of random uniform sampling.*

At larger sample size this benefit diminishes. And since we have shown test distribution (uniform) sampling to be inferior to our proposed sampling scheme, we can already deduce that standard GP uncertainty sampling is inferior, too.

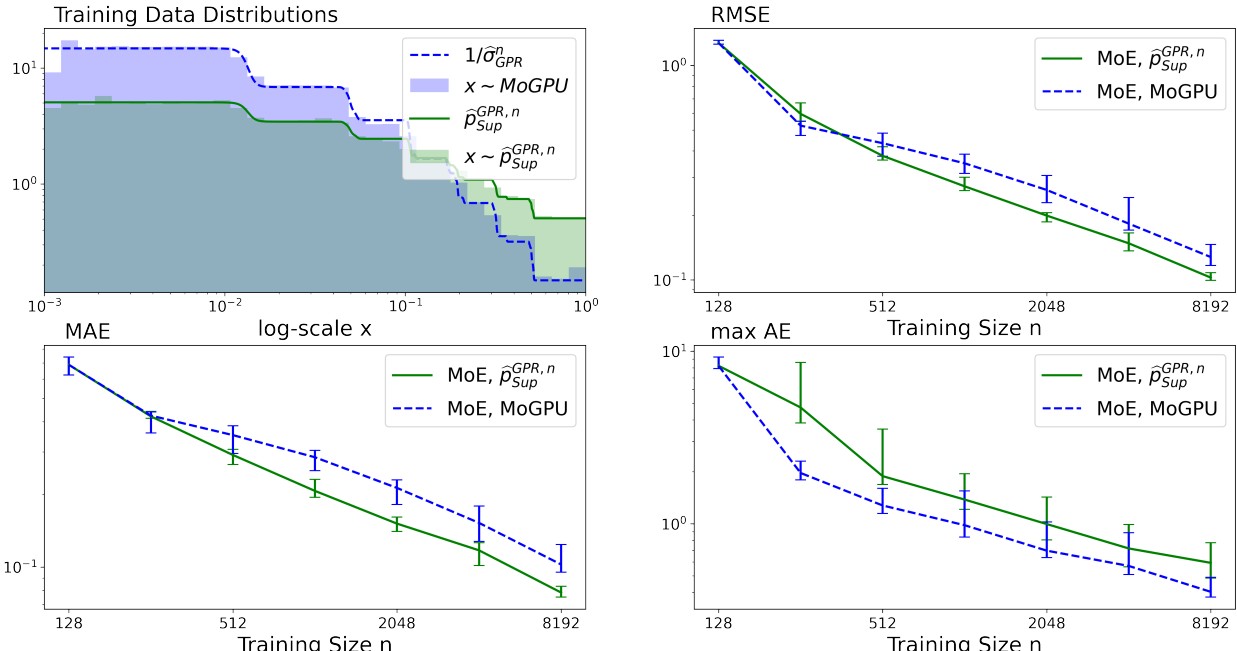

Figure 9: A Comparison of the mixture of Gaussian process uncertainty sampling baseline to our proposed active sampling scheme for the Doppler experiment: (Top left) The training data histograms after $2^{13}$ samples, contrasted with functions of $\sigma_{\mathrm{GPR}}^n$, and RMSE (top right), MAE (bottom left) and the max AE (bottom right) at several training sizes of the compared schemes. The results are averaged over 20 repetitions.

Instead—given the gate function of our MoE from the previous part of this experiment, which was obtained for $2^{15}$ training samples and which we now keep fixed—we implement the proposed MoGPU as defined in (18). Intuitively, the uncertainty estimate in $x \in \mathcal{X}$ increases as the applied bandwidth $\sigma_{\mathrm{GPR}}^n(x)$ decreases. Now, in order to equalize uncertainty over the input space, MoGPU will sample more in regions where $\sigma_{\mathrm{GPR}}^n$ is smaller. For $\boldsymbol{X}_n$ drawn according to MoGPU, we expect $\boldsymbol{X}_n \sim [\sigma_{\mathrm{GPR}}^n]^{-d}$. This expectation holds as can be seen to the upper left in Fig. 9.

For evaluation, we combine the fixed gate function with full GPR experts and compare our proposed sampling scheme with MoGPU (both determined through the gate). In all error measures the beneficial effect of the low-dispersion property of $\boldsymbol{X}_n$ drawn according to MoGPU has already vanished for about 512 training samples, from where the asymptotic law dominates. As expected, our approach is superior to MoGPU when comparing RMSE. Interestingly, MoGPU is superior to our approach regarding the max AE, suggesting that $\boldsymbol{X}_n \sim [\sigma_{\mathrm{GPR}}^n]^{-d}$ is the preferable training distribution under the supremum-norm.

## 5.2 Force field reconstruction

We now turn our attention to a real-world example in which we predict the *potential energy surface* (PES) and corresponding *force field* (FF) of a molecule from first-principles calculations. The PES function links the geometry $x = [R_1, \dots, R_a] \in \mathbb{R}^{3 \times a}$ of a molecule to its potential energy $E \in \mathbb{R}$, where $R_i$ are the Cartesian positions of the $a$ atoms of the molecule. In ab-initio computations, this mapping is achieved by solving the time-independent Schrödinger equation. The PES encodes essential information on the properties of a molecule. Due to thermal and quantum effects, molecules are never perfectly rigid but assume different configurations. The distribution of these configurations is determined by the shape of the PES. For example, minima of the PES will be sampled more frequently than other regions and correspond to stable structures. This has practical implications, since many experimental techniques measure an expectation value over molecular distributions. In order to achieve a meaningful comparison, sampling needs to be taken into account in theoretical simulations as well. One of the most successful approaches to sample molecular distributions are MD simulations. They model the time evolution of the atomic positions, sampling the PES by integrating Newtons equations of motion. To this end, energy conserving forces acting on each atom are required. These forces are the negative derivative of the PES with respect to the atomic positions $F \in \mathbb{R}^{3 \times a}$.

This type of proxy for the prohibitively expensive ab initio quantum mechanical calculations is commonly used to enable long-time scale MD simulations that consist of millions of steps, each requiring the evaluation of the PES and FF for a new geometry. Converged MD trajectories give unique insight into the dynamic behavior and structure-function relationships of physical systems at atomic scale. They are widely used in molecular biology research and play a crucial role in applications such as protein folding and drug discovery. ML has the potential to profoundly advance this field, as it bears the promise of offering a unique cost-accuracy trade-off that is not achievable with traditional methods Noé et al. (2020); von Lilienfeld et al. (2020); Unke et al. (2021b); Keith et al. (2021). However, almost all current ML-based FFs rely on rather naive exhaustive sampling schemes to gather training data, which stands in the way of scaling to larger system sizes, both, from a data acquisition cost and training perspective. Here, we demonstrate how our method can be use to construct smaller, yet more effective training datasets.

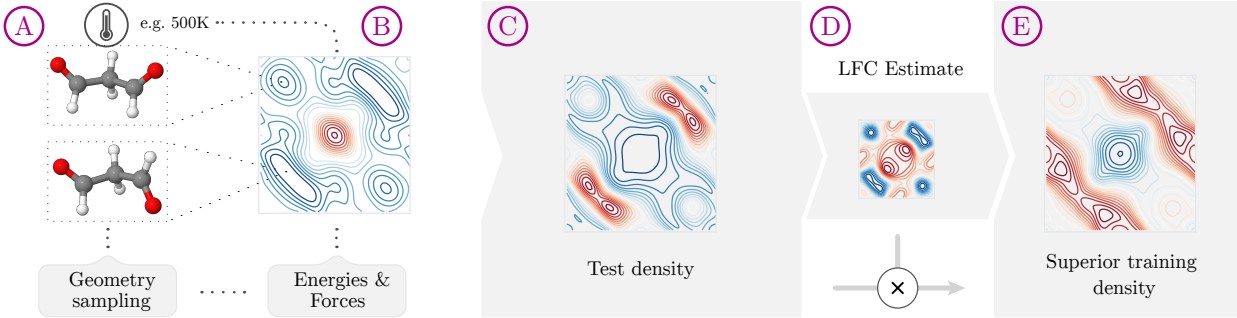

Figure 10: Reconstructing ML-based FFs using our MoE approach: (Left) The inputs and outputs of the regression task are the geometries and energies (including forces, i.e., energy gradients) of malonaldehyde. As an example, we highlight the geometries of the two energetically stable states found in the local minima of the energy surface. (Middle) The density estimate to the true MD geometry distribution. (Right) The *superior training density* estimate (17) based on our approach. All properties are evaluated at the relaxed malonaldehyde configurations and plotted with respect to the angles of the two aldehyde rotors of malonaldehyde (see Chmiela et al. (2018); Sauceda et al. (2020)).

In this experiment, we reconstruct a FF for the molecule malonaldehyde, which has $a = 9$ atoms and the chemical formula $C_3O_2H_4$ (see Fig. 10 (A)). Formally, we try to infer the high-dimensional target function $f : \mathcal{X} \to \mathcal{Y}, R \mapsto [E, F]$, where $\mathcal{X} = \mathbb{R}^{3a}$ and $\mathcal{Y} = \mathbb{R}^{1+3a}$. For visualization purpose we only show a two-dimensional subspace of the PES, which is characterised by the two main features of this molecule, its two rotors (aldehyde groups) (Chmiela et al., 2018; Sauceda et al., 2020). Their relative orientation is the dominant driver of the potential energy in this case and therefore most descriptive. Each point on the surface

depicted in Fig. 10 (B) is generated by fixing the rotor pair at a particular angle and relaxing all remaining degrees of freedom to obtain a minimal energy configuration. We will refer to these geometries as the *relaxed configurations* in the following.

To reconstruct the FF, we use the *symmetric gradient-domain machine learning* (sGDML) FF (Chmiela et al., 2018; 2019), which is a GPR model that takes energy and force labels and also roto-translational and permutational invariances of the geometries into account (see Appendix B.4 for details). We anticipate that sGDML will benefit from our MoE approach, as the transition paths along the PES vary in complexity, due the interplay between distinct atom types with different characteristic interaction length scales. Our active learning approach can only improve training efficiency, if there are inhomogeneities in the data. Using our LFC estimate, we therefore first verify our intuition that the PES of malonaldehyde varies in complexity. Based on this, we derive the *superior training density*, which we finally feed to our active learning framework to refine the training dataset in a superior way.

**Experimental setup** All experiments use an extensive pre-computed reference trajectory (almost a million data points $(X_{\mathrm{MD}}, Y_{\mathrm{MD}})$) as ground truth, as opposed to generating new data points on demand. This test setup allows a post-hoc verification of the training distribution generated by our active learning approach, while still providing ample redundancy and therefore sampling freedom.

Recall from Sec. 3.3 that we require an unnormalized density estimate of the trajectory $X_{\mathrm{MD}} \sim p_{\mathrm{MD}}$, since we are dealing with a pool-based active learning scenario. We estimate $\widehat{p}_{\mathrm{MD}}$ by standard *kernel density estimation*, based on the energy-to-energy entry of the sGDML kernel $\widetilde{\boldsymbol{k}}$ from (31) at $\sigma = 0.03$. Fig. 10 (C) shows the density estimate of the relaxed configurations, where we observe that $p_{\mathrm{MD}}$ is very unbalanced, with strong concentration of mass near the stable configurations.

We implement our MoE approach, using the sGDML kernel $\widetilde{\boldsymbol{k}}$ from (31) with a Gaussian base kernel function $k$. While we sample the training data randomly (with appropriate weights) from the pool, we will draw sub-samples (i.e. for choosing the inducing points of sparse expert and gate models) via *symmetrized DC* with distributional k-means++ initialization (see Appendix C).

Since this dataset comes with practically noise-free labels (we consider the first principle calculations as ground-truth), we tune the experts (and MoE model) with respect to MSE rather than the MLL objective. For stability, we will apply $\widehat{v}(x) = 10^{-9}$ even though we assume no noise.

**Anisotropic bandwidths** sGDML operates on $\boldsymbol{d} = \boldsymbol{a}(\boldsymbol{a}-1)/2 = 36$ features that are based on the interatomic distances of the molecule. In contrast to the work of Chmiela et al. who restrict themselves to an isotropic bandwidth $\Sigma_E = \sigma_E \mathcal{I}_{\boldsymbol{d}}$, our implementation of sGDML in GPyTorch naturally enables us to tune an anisotropic bandwidth $\Sigma_E = \boldsymbol{diag}(\sigma_1, \ldots, \sigma_{\boldsymbol{d}})$ in the preprocessing step.

We partially offset the increased memory footprint of the model due to the tunable $\Sigma_E$ by implementing the sparse GPR model from Appendix B.2 under the sGDML kernel $\widetilde{\boldsymbol{k}}$ from (31) and limiting the number of inducing points to $m = 128$ configurations. Since all our features are of the same type—pairwise interatomic distances—they are inherently calibrated in terms of scale. Hence, the reciprocal entries of $\Sigma_E$ directly translate into importance of the features, which we display in Fig. 11.

We observe, that the importance assigned to some pairs of atoms agree with chemical intuition, e.g., interactions with light hydrogen atoms are generally weaker. Furthermore, the important role of the opposing aldehyde groups in malonaldehyde emerges in the form of a heavily weighted path that connects the O-C-C-C-O backbone of the molecule. Unsurprisingly, the dataset also yields informative spurious atomic interactions, that are not chemically obvious.

In Fig. 12 we see that our anisotropic variant of sGDML performs consistently better than the original isotropic sGDML model. Similar to the calculation of the relative sample size in (19) we can compare two models of equal asymptotic MSE law. When comparing anisotropic to isotropic sGDML, both under *random test sampling*, we can save about 10% of samples.

**Setting up the MoE model** After having trained $\Sigma_E$, we apply dense sGDML experts with $\Sigma_j = \sigma_j \Sigma_E$, where $\sigma_j = 2^{-5/4+j/2}, 1 \le j \le 8$ as the individual expert bandwidths, $\lambda_E = 1$ as the initial expert regularization, and $\sigma_G = 0.1$ and $\lambda_G = 10^4$ for the sparse gate with 1024 inducing points. For the training,

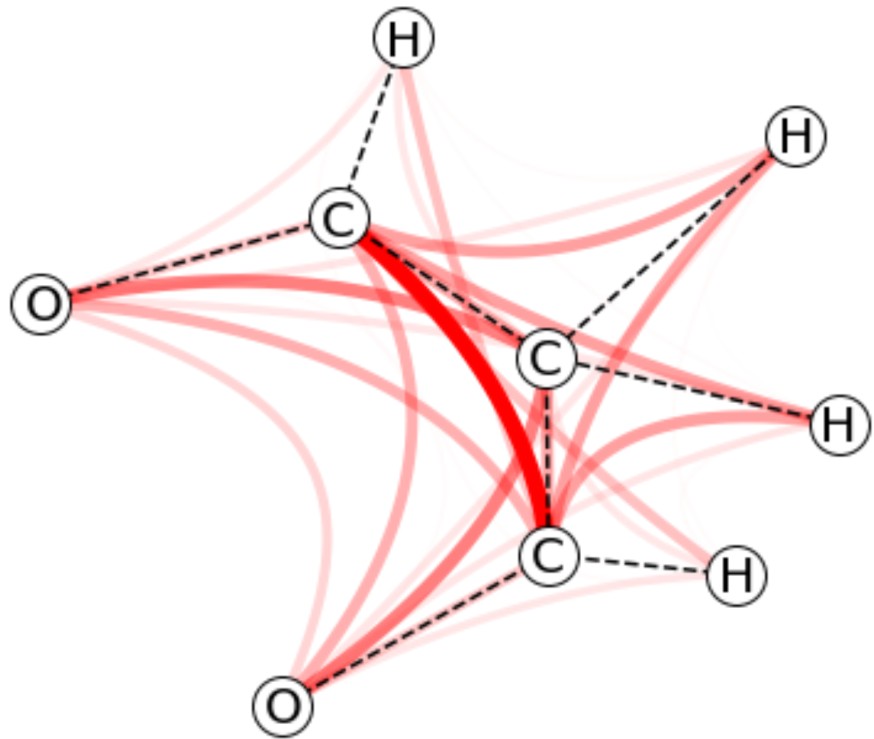

Figure 11: A visualization of the individual feature importance of malonaldehyde in the anisotropic case: The structural formula of the molecule is plotted in black. The importance of the individual interatomic distances is reciprocal to $\Sigma_E$, which is the bandwidth estimate obtained by training the anisotropic sGDML model. Hence, we express the importance of each interatomic distance of the molecule in red, where the importance corresponds to the line saturation $\gamma = [[-\log(\Sigma_E) - \min(-\log(\Sigma_E))]/[\max(-\log(\Sigma_E)) - \min(-\log(\Sigma_E))]]^2$.

we apply a batch size of $B = 1024$, a terminal expert sparsity $\kappa = 8$, a penalty factor of $\vartheta_\sigma = 0.01$ for small bandwidth choices, gate noise parameters $\mathfrak{s}_0 = 0.01$ and $\eta_{\mathfrak{s}} = 1/\sqrt{2}$, and learning rate parameters $\eta = 0.005$, $\eta_H = 0.05$, $\eta_G = 0.1$. Like we will discuss in Appendix D, for tuning the MoE with dense sGDML experts, we either require an additional training set, which is independent of the training set for the experts, or we could provide leave-one-out (LOO) responses of the experts for the training of the gate. In our experiment, we use an additional gate training set $\boldsymbol{X}_{n_G}^G$ of fixed size $n_G = 2^{14}$. The anisotropic MoE model performs consistently better than anisotropic sGDML, as can be seen in Fig. 12. When comparing the anisotropic MoE model to isotropic sGDML, both under *random test sampling*, we can save about 21% of samples.

**Active learning**  We assume an intrinsic dimension of $d = 2$ (the two aldehyde rotor angles, the most salient features of malonaldehyde) and a smooth target function $f \in \mathcal{C}^\infty(\mathcal{X}, \mathcal{Y})$. The test distribution is given by the MD trajectory such that $q = p_{\mathrm{MD}}$. Prior to the active learning procedure, we separate the validation samples $x_{\mathrm{val}}$ and test samples $x_{\mathrm{test}}$ at random from the pool $X_{\mathrm{MD}}$. We apply an initial expert training size of $n_0 = 2^9$, doubling the sample size with each iteration of the active learning procedure. The initial expert training set $\boldsymbol{X}_{n_0}$ and the gate training set $\boldsymbol{X}_{n_G}^G$ are drawn via importance sampling from the remaining pool with weights $\widehat{p}_{\mathrm{MD}}^{-1/2}(X_{\mathrm{MD}} \setminus (x_{\mathrm{val}} \cup x_{\mathrm{test}}))$. By this it is $\boldsymbol{X}_{n_0} \sim q^{1/2}$, which is more in alignment with the *superior training density* (17) than sampling $\boldsymbol{X}_{n_0} \sim q$.

In Fig. 10 (D, E) we show the estimates to LFC and the *superior training density* under the pool test distribution, evaluated on the relaxed configurations of malonaldehyde. The LFC estimates confirm our expectation that the transition areas are more complex to model than the regions near the stable configu-

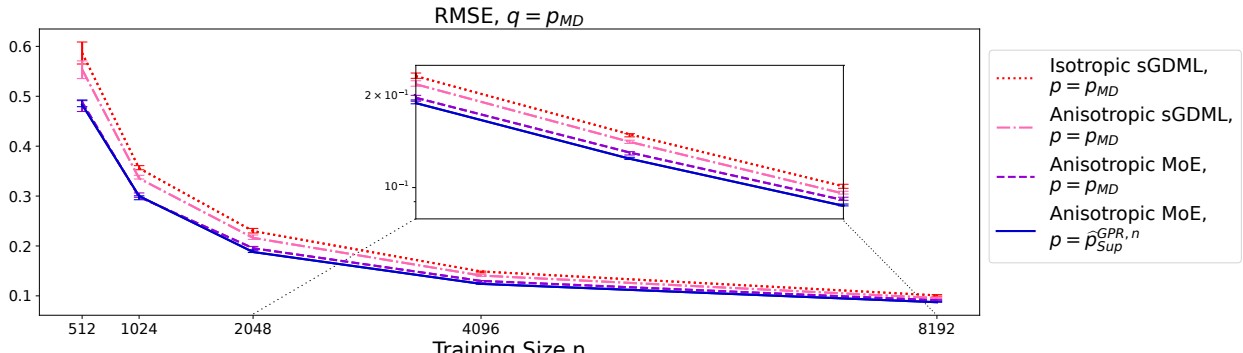

Figure 12: The RMSE under the true MD trajectory test distribution for different variants of sGDML and training distribution at varying training size: The performance is given for passive sampling, using the original isotropic sGDML (dotted), anisotropic sGDML (dash-dotted) and our MoE model with anisotropic sGDML experts (dashed), and for the MoE model, applying the proposed active learning framework (solid). The results are averaged over 5 repetitions.

rations. Subsequently, our active sampling scheme shifts sample mass away from the stable regimes in favor of the transition areas.

We have plotted the error curves of passive and active sampling schemes in Fig. 12. When estimating the relative sample size (19) that we require to achieve the same RMSE via active sampling compared to *random test sampling*, we obtain $\varrho(\widehat{f}_{\mathrm{MoE}}, \widehat{p}_{\mathrm{Sup}}^{\mathrm{GPR},n}) = 0.920 \pm 0.013$. This means that we save about 8% of samples under the MoE model with our active sampling scheme compared to *random test sampling*. In total, when comparing our actively trained MoE approach to the passively trained, original sGDML model, we can save about 31% of samples. Notably, DFT level calculations (Perdew et al., 1996; Blum et al., 2009; Tkatchenko & Scheffler, 2009) for the studied system require minutes to hours of computation *per sample*, CCSD(T) level computations even require days of computation per sample. So in the field of quantum chemistry saving roughly a third of computing power is of practical importance.

## 6 Conclusion

Standard machine learning tasks implicitly assume a certain homogeneity in the data scales. However in practice this structural property of the learning problem may not be fulfilled, e.g., in multiscale problems from the sciences such as turbulence Brunton et al. (2020) or quantum chemistry Noé et al. (2020); von Lilienfeld et al. (2020); Unke et al. (2021b); Keith et al. (2021).

In this work, we aimed at identifying local inhomogeneities in regression tasks, which can be used to construct better models and training datasets and for domain interpretation. To this end, we combined recent results on model-agnostic LFC estimates and asymptotically optimal sampling, which are founded in the domain of LPS, with estimates to LOB, which are derived in the GPR domain. By this, we benefit from both sides, having a theoretically sound optimal sampling scheme on the on hand, and having access to the required estimates from a model that naturally can cope with high input space dimensions on the other hand.

On synthetic data, we showcased and validated our approach, where we analyzed similarities with the LPS-based analog, but also compared to the most related GP uncertainty sampling concepts for active learning. To show the full potential of our approach, we studied a real-world, high-dimensional force field reconstruction task. Here, we first identified the multi-scale structure of the problem, whose treatment also reflects in a substantial performance gain of the broadly adopted method sGDML. Our approach then not only gave access to an interpretable visualization of this inhomogeneity, but also guided the sampling process in a way that takes the structural changes into account, enhancing the quality of the training data. Future work will include the investigation of sparse Gaussian process models that analytically solve for the inducing values (Seeger et al., 2003; Titsias, 2009) as experts for our MoE model. This would reduce the number of parameters and, hence, the computational complexity of our approach. Furthermore, we will focus on

the application of our approach to real-world problems from chemistry, physics and further domains also applying techniques from *explainable AI* (e.g. Samek et al. (2021); Letzgus et al. (2022)). In particular, recent advances on sGDML regarding the scalability by Chmiela et al. (2022) will enable the application of our approach to large molecular systems.

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

# A  Asymptotic results for local polynomial smoothing

In this section, we will review the theory of Panknin et al. (2021).

The prediction of the LPS model of order $Q$ under the bandwidth $\Sigma \in \mathbb{S}_{++}^d$ in $x \in \mathcal{X}$ can be understood as follows: First, the the regression problem is localized around $x$ according to weights $k^\Sigma(\cdot, x)$ that decrease with growing distance to $x$. Then we search for the polynomial up to order $Q$ that fits the localized regression problem best. Finally, the evaluation of this polynomial in $x$ is returned as the prediction. Formally, it is

$$m_Q^\Sigma(x) = \mathfrak{p}_{Q,\Sigma,x}^*(0), \text{ where} \tag{20}$$
$$\mathfrak{p}_{Q,\Sigma,x}^* = \underset{\mathfrak{p} \in \mathcal{P}_Q(\mathbb{R}^d)}{\arg\min} \sum_{i=1}^n k^\Sigma(x_i, x) \left(y_i - \mathfrak{p}(x_i - x)\right)^2,$$

and $\mathcal{P}_Q(\mathbb{R}^d)$ is the space of the real polynomial mappings $\mathfrak{p} \colon \mathbb{R}^d \to \mathbb{R}$ up to order $Q$.

The localization is controlled by $\Sigma$ through the kernel weights $\boldsymbol{K}^\Sigma(x, x_i)$ for $x_i \in \boldsymbol{X}_n$: For an RBF-kernel, $k^\Sigma(x, x')$ decays monotonically with growing distance of $x'$ to $x$. This decay is dampened or amplified as $\Sigma$ increases, respectively decreases (in the sense of the Loewner order).

For readability, since $\Sigma$ will be replaced by terms with more involved notation, we redefine (1) by

$$\text{MSE}\left(x, \widehat{f}, \Sigma | \boldsymbol{X}_n\right) := \text{MSE}\left(x, \widehat{f}^\Sigma | \boldsymbol{X}_n\right). \tag{21}$$

For a bandwidth space $\mathcal{S} \subseteq \mathbb{S}_{++}^d$, Panknin et al. (2021) proposed to minimize the active learning objective

$$\text{MISE}\left(q, \widehat{f} | \boldsymbol{X}_n\right) = \int_{\mathcal{X}} \inf_{\Sigma \in \mathcal{S}} \text{MSE}\left(x, \widehat{f}, \Sigma | \boldsymbol{X}_n\right) q(x) dx, \tag{22}$$

which is the optimal *mean integrated squared error*, obtained by predictions that are based on locally optimal chosen bandwidths. If these locally optimal bandwidth choices are well-defined, that is, if for all $x \in \mathcal{X}$ there exists a unique $\Sigma' \in \mathcal{S}$ such that

$$\text{MSE}\left(x, \widehat{f}, \Sigma' | \boldsymbol{X}_n\right) = \inf_{\Sigma \in \mathcal{S}} \text{MSE}\left(x, \widehat{f}, \Sigma | \boldsymbol{X}_n\right),$$

we are able to define the *locally optimal bandwidth* (LOB) function

$$\Sigma^n(x) = \mathbf{arg\,min}_{\Sigma \in \mathcal{S}} \text{MSE}\left(x, \widehat{f}, \Sigma | \boldsymbol{X}_n\right).$$

This function exists, for example, in the *isotropic* case $\mathcal{S} = \{\sigma \mathcal{I}_d \mid \sigma > 0\}$ for LPS under mild conditions, where we denote $\Sigma^n(x) = \sigma^n(x)\mathcal{I}_d$ (see, e.g., Masry (1996; 1997); Fan et al. (1997) or Panknin et al. (2021) for an overview).

Assuming the isotropic bandwidths candidate space $\mathcal{S} = \{\sigma \mathcal{I}_d \mid \sigma > 0\}$, the LOB as in Eq. (4) is an asymptotically well-defined function under mild assumptions[3]: Denoting the LOB of LPS of order $Q$ by $\Sigma_Q^n(x) = \sigma_Q^n(x)\mathcal{I}_d$ such that

$$\sigma_Q^n(x) = \mathbf{arg\,min}_{\sigma>0} \text{MSE}\left(x, m_Q, \sigma \mathcal{I}_d | \boldsymbol{X}_n\right), \tag{23}$$

asymptotically it holds

$$\sigma_Q^n(x) = C_Q \left[\frac{v(x)}{p(x)n}\right]^{\frac{1}{2(Q+1)+d}} \mathrm{b}_Q\left[x, \mathcal{I}_d\right]^{-\frac{2}{2(Q+1)+d}} + o_p\left[n^{-\frac{1}{2(Q+1)+d}}\right], \tag{24}$$

where $C_Q$ is a constant, and $\mathrm{b}_Q\left[x, \mathcal{I}_d\right]$ is a function of $x$ taken from the asymptotic *conditional bias* $f(x) - \mathbb{E}\left[m_Q^{h_n \mathcal{I}_d}(x)\Big| \boldsymbol{X}_n\right]$ of LPS (Masry, 1996; 1997). That is, for a sequence $h_n \to 0$ as $n \to \infty$ we can write the conditional bias, which is of order $Q+1$, as

$$f(x) - \mathbb{E}\left[m_Q^{h_n \mathcal{I}_d}(x)\Big| \boldsymbol{X}_n\right] = h_n^{Q+1}\mathrm{b}_Q\left[x, \mathcal{I}_d\right] + o_p\left[h_n^{Q+1}\right]. \tag{25}$$

Eq. (24) shows how LOB scales asymptotically with respect to the training size $n$, the local noise level function $v(x)$ and the training density $p(x)$. The remaining bias-component depends on the local structural complexity, which can be characterized by the derivatives of $f$ in a non-trivial way. Therefore it encodes the local structural complexity of $f$. Given all other properties and LOB itself, we are able to formulate LFC in a closed form.

**Definition 4** (Panknin et al. (2021))**.** *For LPS of order $Q$, the LFC of $f$ in $x \in \mathcal{X}$ is asymptotically given by*

$$\mathfrak{C}_Q^n(x) = \left[\frac{v(x)}{p(x)n}\right]^{\frac{d}{2(Q+1)+d}} \left|\Sigma_Q^n(x)\right|^{-1} = \left[\frac{v(x)}{p(x)n}\right]^{\frac{d}{2(Q+1)+d}} \sigma_Q^n(x)^{-d}.$$

As already mentioned in Eq. (22), given a test density $q$, the active learning task is to minimize $\text{MISE}\left(q, m_Q | \boldsymbol{X}_n\right)$. Now, if LOB is well-defined, we can rewrite

$$\text{MISE}\left(q, m_Q | \boldsymbol{X}_n\right) = \int_{\mathcal{X}} \inf_{\Sigma \in \mathcal{S}} \text{MSE}\left(x, m_Q, \Sigma | \boldsymbol{X}_n\right) q(x)dx$$

$$= \int_{\mathcal{X}} \text{MSE}\left(x, m_Q, \Sigma_Q^n(x) | \boldsymbol{X}_n\right) q(x)dx.$$

Finally, when solving for the optimal training dataset

$$\boldsymbol{X}_n' \approx \mathbf{arg\,min}_{\boldsymbol{X}_n \in \mathcal{X}^n} \text{MISE}\left(q, m_Q | \boldsymbol{X}_n\right),$$

as in Eq. (3), the optimal training inputs $\boldsymbol{X}_n'$ can be written asymptotically as an independent and identically distributed sample from the optimal training distribution, whose density $p_{\text{Opt}}^{Q,n}$ possesses an asymptotic closed form.

---

[3]We require non-vanishing leading bias- and variance-terms of $m_Q(x)$, which is guaranteed if $\forall x \in \mathcal{X}$ it holds that $\mathrm{b}_Q\left[x, \mathcal{I}_d\right] \neq 0$ from Eq. (25) and $v(x) > 0$.

**Theorem 5** (Panknin et al. (2021)). *Let $v, q \in \mathcal{C}^0\left(\mathcal{X}, \mathbb{R}_+\right)$ for a compact input space $\mathcal{X}$, where $q$ is a test probability density. Additionally, assume that $v$ and $q$ are bounded away from zero. I.e., $v, q \geq \epsilon$ for some $\epsilon > 0$. Let $k$ be a RBF-kernel with bandwidth parameter space $\mathcal{S} = \{\sigma \mathcal{I}_d \mid \sigma > 0\}$. Let $Q \in \mathbb{N}$ be odd and $f \in \mathcal{C}^{Q+1}(\mathcal{X})$ such that the bias of order $Q+1$ does not vanish almost everywhere. Then the optimal training density for LPS of order $Q$ is asymptotically given by*

$$p_{Opt}^{Q,n}(x) \propto \left[\mathfrak{C}_Q^n(x)q(x)\right]^{\frac{2(Q+1)+d}{4(Q+1)+d}} v(x)^{\frac{2(Q+1)}{4(Q+1)+d}}(1 + o(1)).$$

We will use this optimal distribution to sample $\boldsymbol{X}_n' \sim p_{\mathrm{Opt}}^{Q,n}$ with a proposed estimator for $\mathfrak{C}_Q^n$ that is scalable with respect to the input space dimension.

For LPS with $\boldsymbol{X}_n' \sim p$ and $\boldsymbol{X}_n \sim q$, we can asymptotically calculate the relative required sample size from Definition 2 in Sec. 3.2 by

$$\varrho(m_Q, p) = \left[\frac{\mathrm{MISE}\left(q, m_Q | \boldsymbol{X}_n'\right)}{\mathrm{MISE}\left(q, m_Q | \boldsymbol{X}_n\right)}\right]^{\frac{2(Q+1)+d}{2(Q+1)}}. \tag{26}$$

# B  Preliminaries on the applied models

In this section, we give further details on all the models that we implement in this work. For the RBF-kernel $k$, we define the kernel matrix between $X \in \mathcal{X}^n$ and $X' \in \mathcal{X}^m$ as $\boldsymbol{K}^\Sigma(X, X') = \left[k^\Sigma(x, x')\right]_{x \in X, x' \in X'}$. As a shorthand notation we furthermore define $\boldsymbol{K}^\Sigma(X) := \boldsymbol{K}^\Sigma(X, X)$.

## B.1  Gaussian process regression

The GPR model $\widehat{y} \sim \mathcal{GP}(\theta)$ (see, e.g. Williams & Rasmussen (1996)) is defined as follows: The Gaussian process is described by the hyperparameters $\theta = (\mu, \lambda, \widehat{v}, \Sigma)$, which are the global constant prior mean $\mu$, the regularization parameter $\lambda$, the label noise variance function $\widehat{v}$ and the bandwidth matrix $\Sigma$ of the kernel. If we can assume homoscedastic noise, we let $\widehat{v}(x) \equiv \sigma_\varepsilon^2$.

The prior Gaussian process then assumes the labels $\boldsymbol{Y}_n$ of $\boldsymbol{X}_n$ to be distributed according to $\boldsymbol{Y}_n = \widehat{y}(\boldsymbol{X}_n) \sim \mathcal{N}(\cdot; \mu(\boldsymbol{X}_n), \boldsymbol{C}(\boldsymbol{X}_n)|\theta)$, for the constant mean function $\mu(\boldsymbol{X}_n) = \mu \mathbb{1}_n$, and the covariance function

$$\boldsymbol{C}(\boldsymbol{X}_n) = \lambda \boldsymbol{K}_n + \boldsymbol{diag}(\widehat{v}(\boldsymbol{X}_n)),$$

where $\boldsymbol{K}_n = \boldsymbol{K}^\Sigma(\boldsymbol{X}_n)$ is the kernel matrix of $\boldsymbol{X}_n$.

For test inputs $\boldsymbol{X}_*$, the posterior predictive distribution of $\boldsymbol{Y}_*$ is then given by

$$\widehat{y}(\boldsymbol{X}_*) \sim \mathcal{N}(\cdot; \mu^*(\boldsymbol{X}_*), \boldsymbol{C}^*(\boldsymbol{X}_*)|\theta),$$

where the predictive mean and covariance are given by

$$\mu^*(\boldsymbol{X}_*) = \mu(\boldsymbol{X}_*) + \boldsymbol{C}_{*n}\boldsymbol{C}_n^{-1}(\boldsymbol{Y}_n - \mu(\boldsymbol{X}_n)), \tag{27}$$

$$\boldsymbol{C}^*(\boldsymbol{X}_*) = \boldsymbol{C}_* - \boldsymbol{C}_{*n}\boldsymbol{C}_n^{-1}\boldsymbol{C}_{*n}^\top, \tag{28}$$

and we have defined

$$\boldsymbol{C}(\boldsymbol{X}_n \cup \boldsymbol{X}_*) = \begin{bmatrix} \boldsymbol{C}_n & \boldsymbol{C}_{*n}^\top \\ \boldsymbol{C}_{*n} & \boldsymbol{C}_* \end{bmatrix}.$$

Note that test predictions $\widehat{f}_{\mathrm{GP}}(x) = \mu^*(x)$ are given by Eq. (27).

## B.2  Sparse Gaussian processes

Let us define the sparse GPR model $\widehat{y} \sim \mathcal{SGP}(\theta)$ (see, e.g. Williams & Rasmussen (1996); Hensman et al. (2015)) as follows: The sparse Gaussian process is described by the (hyper-) parameters

$\theta = (\mu, \lambda, \widehat{v}, \Sigma, \boldsymbol{X}^*, \boldsymbol{\mu}^*, \boldsymbol{S}^*)$, which are the global constant prior mean $\mu$, the regularization parameter $\lambda$, the label noise variance function $\widehat{v}$, the bandwidth matrix $\Sigma$ of the kernel and the prior distribution, given by the inducing point locations $\boldsymbol{X}^* \in \mathcal{X}^m$ as well as their inducing value distribution, characterized by the moments $\boldsymbol{\mu}^*$ and $\boldsymbol{S}^*$. That is, for the inducing values $\boldsymbol{Y}^*$ of $\boldsymbol{X}^*$ we assume $\boldsymbol{Y}^* = \widehat{y}(\boldsymbol{X}^*) \sim \mathcal{N}(\cdot; \boldsymbol{\mu}^*, \boldsymbol{S}^*)$. Here, the degree of sparsity is described by $m$ inducing points: This number can be fixed in advance or gradually increased with training size $n$, where the increase $m_n = o[n]$ is typically much slower than $n$. If we can assume homoscedastic noise, we let $\widehat{v}(x) \equiv \sigma_\varepsilon^2$.

The sparse Gaussian process then outputs

$$\widehat{y}(\boldsymbol{X}_n) \sim \mathcal{N}(\cdot; \mu(\boldsymbol{X}_n), \boldsymbol{C}(\boldsymbol{X}_n) | \theta_e)$$

for the mean function

$$\mu(\boldsymbol{X}_n) = \boldsymbol{K}_{n*} \boldsymbol{K}_*^{-1/2} (\widetilde{\boldsymbol{\mu}}^* - \boldsymbol{K}_*^{-1/2} \bar{\mu}(\boldsymbol{X}^*)) + \bar{\mu}(\boldsymbol{X}_n),$$

and the covariance function

$$\boldsymbol{C}(\boldsymbol{X}_n) = \lambda \left[ \boldsymbol{K}_n + \boldsymbol{K}_{n*} \boldsymbol{K}_*^{-1/2} (\widetilde{\boldsymbol{S}}^* - \mathcal{I}_m) \boldsymbol{K}_*^{-1/2} \boldsymbol{K}_{n*}^\top \right] + \boldsymbol{diag}(\widehat{v}(\boldsymbol{X}_n)),$$

where $\widetilde{\boldsymbol{\mu}}^* = \boldsymbol{K}_*^{-1/2} \boldsymbol{\mu}^*$ and $\widetilde{\boldsymbol{S}}^* = \boldsymbol{K}_*^{-1/2} \boldsymbol{S}^* \boldsymbol{K}_*^{-1/2}$ are the whitened moments of the inducing value distribution (Pleiss et al. (2020), Sec. 5.1), and we have defined $\boldsymbol{K}_n = \boldsymbol{K}^\Sigma(\boldsymbol{X}_n)$, $\boldsymbol{K}_* = \boldsymbol{K}^\Sigma(\boldsymbol{X}^*)$ and $\boldsymbol{K}_{n*} = \boldsymbol{K}^\Sigma(\boldsymbol{X}_n, \boldsymbol{X}^*)$.

We choose $\bar{\mu}$ to be the constant mean function, i.e. $\bar{\mu}(X) = \mu \mathbb{1}_n$ for $X \in \mathcal{X}^n$, noting that other mean functions are possible.

### B.3 Sparse mixture of experts

Given a finite set of expert models $\widehat{y}_l$ that are parameterized by $\theta_{e_l}$, the mixture of experts (MoE) model is given by

$$\widehat{f}_{\text{MoE}}(x) = \sum_{l=1}^{L} G(x)_l \widehat{y}_l(x),$$

where the *gate* $G : \mathcal{X} \to [0,1]^L$ is a probability assignment of an input $x$ to the experts. In particular, it holds $\sum_{l=1}^{L} G(x)_l \equiv 1$ and $G(x)_i \geq 0, \forall x \in \mathcal{X}$ and $1 \leq i \leq L$.

We implement the approach of Shazeer et al. (2017) to model the gate $G$ as follows: Let us define the *softmax* function of $\boldsymbol{a} \in \mathbb{R}^L$ as

$$\mathbf{soft\,max}(\boldsymbol{a})_i = \exp\{\boldsymbol{a}_i\} \Big/ \sum_{l=1}^{L} \exp\{\boldsymbol{a}_l\}. \tag{29}$$

Shazeer et al. (2017) propose to set

$$G(x) = \mathbf{soft\,max}(\widetilde{h}_1(x), \ldots, \widetilde{h}_L(x)), \quad \text{where} \quad \widetilde{h}_{i_l}(x) = \begin{cases} h_{i_l}(x) & , l < \kappa \\ -\infty & , l \geq \kappa + 1 \end{cases} \tag{30}$$

for an adequate permutation $(i_1, \ldots, i_L)$ of $\{1, \ldots, L\}$ such that $h_{i_l}(x) > h_{i_{l+1}}(x)$ are ordered decreasingly. Here, $h_l(x) = g_l(x) + \mathcal{N}(0, \mathfrak{s}_l^2)$ is a noisy version of single-channel gating models $g_l$ with parameters $\theta_{g_l}$. Note that these models can be chosen freely and may also deviate from the choice of expert models $\widehat{y}_l$.

The cutoff value $1 \leq \kappa \leq L$ controls the sparsity of the MoE, as it enforces the minor mixture weights to strictly equal zero. For stability reasons, while training, we give each expert a chance to become an element of the top-$\kappa$ components by adding independent Gaussian noise $\mathcal{N}(0, \mathfrak{s}_l^2)$ before thresholding, where $\mathfrak{s} \in \mathbb{R}_{++}^L$ is another hyperparameter to set or learn. This noisy gating prevents a premature discarding of initially underperforming experts.

The overall MoE hyperparameter set is thus given by

$$\Theta = (\{\theta_{e_l}\}_{l=1}^L, \{\theta_{g_l}\}_{l=1}^L, \kappa, \mathfrak{s}).$$

### B.4 The sGDML model

The GDML model by Chmiela et al. (2017) represents the geometry $x = [R_1, \ldots, R_a] \in \mathbb{R}^{3 \times a}$ of each molecule in terms of the reciprocal distances $\Phi(x)_{kl} = \|R_k - R_l\|^{-1}$ of all atom pairings to achieve roto-translational invariance of the input. This representation gives us a total $d = a(a-1)/2$ input features. The similarity of a pair of configurations $(z, E, F)$ and $(z', E', F')$ is then given by the extended covariance function

$$\boldsymbol{Cov}(E, E') = k(\Phi(z), \Phi(z')),$$
$$\boldsymbol{Cov}(E, F') = \frac{dk(\Phi(z), \Phi(z'))}{d\Phi'} \frac{d\Phi(z')}{dx},$$
$$\boldsymbol{Cov}(F, F') = \left[\frac{d\Phi(z)}{dx}\right]^\top \frac{dk(\Phi(z), \Phi(z'))}{d\Phi d\Phi'} \frac{d\Phi(z')}{dx}.$$

Hence, we denote the overall kernel function of two configurations by

$$\boldsymbol{k}(z, z') = \boldsymbol{Cov}((E, F), (E', F')) \in \mathbb{R}^{(3a+1) \times (3a+1)}.$$

Atoms of the same type are physically identical and therefore exchangeable, albeit only a small subset of such symmetries is exercised at a given (low) MD simulation temperature. Full permutational invariance is only needed, when enough energy is put into the system for all bonds to break and all atoms to disassociate.

The symmetric extension sGDML Chmiela et al. (2018; 2019) automatically identifies all accessed atom permutations from the training set and adds this symmetric prior to the covariance function. Formally, let $(\pi_s)_{s=1}^s$ be atomic permutations that lead to an equivalent molecular representation. Then, the extended symmetric kernel of sGDML is given by

$$\widetilde{\boldsymbol{k}}(z, z') = \sum_{s=1}^s \sum_{t=1}^s \boldsymbol{k}(\pi_s z, \pi_t z'). \tag{31}$$

Malonaldehyde possesses $s = 4$ such permutations.

**Remark 6.** *The identified set of permutations is transitively closed to form a group. Under isotropy, it suffices to permute only one of the two configurations given to the kernel: Permuting both entries (as in 31) equals permuting one entry and multiplying by the constant $s$. However, if the applied bandwidth is not of the form $\Sigma = \sigma \mathcal{I}_d$, this property does not hold.*

## C   Initializing the inducing point locations

Since the number of inducing points of the gate $\boldsymbol{X}_G^*$, as well as the experts $\boldsymbol{X}_E^*$ are the computational bottleneck of our model, they should be chosen advisedly. Let us interpret the choice of inducing points as a nested AL task at small sample size. In the small sample size regime, input space geometric arguments have proven to be robust and superior in comparison to naive approaches like random sub-sampling from the training inputs (Teytaud et al., 2007; Yu & Kim, 2010; Wu, 2019; Liu et al., 2021). They are representative, respecting the training distribution, and diverse (with low-dispersion) so that they achieve an acceptable representation of the dataset at the smallest possible number of inducing points. Indeed, by sampling the inducing points in this manner, we can reduce the distance of an evaluation point $x$ to its closest neighbor in $\boldsymbol{X}_E^*$, which is known to reduce the reconstruction error of a full kernel matrix by a sparse representation (Zhang et al., 2008), as also discussed in Sec. 4.

In addition, recall that our derived LFC measure of local structural complexity quantifies the local variation of the target function. Intuitively it is clear that we require more inducing points to sense and reconstruct the target function where this local variation is higher. In summary, we therefore propose to choose the inducing points $\boldsymbol{X}_E^*, \boldsymbol{X}_G^* \sim \sqrt{p \cdot \mathfrak{C}_{\text{GPR}}^n}$ in a diverse way, respecting LFC and the training density $\boldsymbol{X}_n \sim p$.

In order to obtain diverse inducing point locations with a certain distribution, we consider two approaches, *Stein variational gradient descent* (SVGD) (Liu & Wang, 2016; Han & Liu, 2018) and *distributional clustering* (DC) (Krishna et al., 2019).

**Stein Variational Gradient Descent**  SVGD takes a particle swarm and tries to align the empirical distribution of the particles with a target distribution, of which we require the density, as well as its derivative (Liu & Wang, 2016). In addition, the individual particles repel each other, such that we have both diversity and representativeness. In our scenario we have no access to this derivative, such that we resort to the work of Han & Liu (2018) that is solely based on the density. Since the particles move freely in the input space and we have to evaluate the target density a considerable number of times, we suggest to apply SVGD, when we deal with well-behaved input spaces and target densities that are easy to evaluate. If the input spaces is only given through high-dimensional features from a finite set of samples, SVGD might move particles into regions far apart from the data manifold.

**Distributional Clustering**  DC is similar to the known *k-means clustering* (Gan et al., 2020), but solves a different *inertia* objective, that is modified such that asymptotically, as the number of cluster centers $|\boldsymbol{X}^*| \to \infty$, the distribution of the training data is preserved (Krishna et al., 2019). Under the standard k-means clustering objective, we would observe $\boldsymbol{X}^* \sim p^{\frac{d}{2+d}}$ (Graf & Luschgy, 2007), where it was $\boldsymbol{X}_n \sim p$. Since we intend to use clustering for sub-sampling rather than identifying a fixed number of true cluster centers, we deal with a comparably large number of cluster centers, here. Thus, we will use DC so as to obtain a representative set of inducing points. Due to very mild assumptions on the problem, DC is specifically easy to perform in higher dimensions.

**Dealing with local optima of DC**  The inertia objective of DC is given by

$$\text{inertia}_{\text{DC}}[\boldsymbol{c}|\boldsymbol{X}_n] = \sum\nolimits_{c \in \boldsymbol{c}} \sum\nolimits_{x \in I_c} \mathbb{1}_{x \neq c} \log \|x - c\|, \tag{32}$$

where

$$I_c = \left\{ x \in \boldsymbol{X}_n \mid \|x - c\| \leq \|x - c'\|, \forall c' \in \boldsymbol{c} \right\} \tag{33}$$

are those elements in $\boldsymbol{X}_n$ that are closest to the center $c$.

In the classical Lloyd-step the centers are updated so as to minimize the intra-cluster inertia, which is given in the case of DC by

$$c^* = \arg\min\nolimits_{z \in I_c} \sum\nolimits_{x \in I_c} \mathbb{1}_{x \neq z} \log \|x - z\|. \tag{34}$$

It is a known problem that k-means-related inertia objectives suffer from local optima Arthur & Vassilvitskii (2007): The converged solution of cluster centers will typically lie close to their initialization. One way to tackle this issue in practice is to run multiple repetitions of the procedure, followed by choosing the solution with minimal inertia. Unfortunately, the amount of local optima increases with the number of cluster centers. And in our case, where we use DC for sub-sampling rather than clustering in its usual sense, we deal with a large amount of clusters such that this strategy becomes computationally tedious.

Complementary to running multiple repetitions of k-means, we will extend the state-of-the-art method *k-means++* for choosing the initial set of clusters in a more sophisticated way, where we additionally account for the training distribution. Given the inertia objective

$$\text{inertia}[\boldsymbol{c}|\boldsymbol{X}_n] = \sum\nolimits_{i=1}^{n} \min\nolimits_{c \in \boldsymbol{c}} \|x_i - c\|^2$$

of the cluster centers $\boldsymbol{c}$, the *k-means++* procedure builds the set of initial cluster centers as follows: Draw the first center $c_1$ randomly from $\boldsymbol{X}_n$. Then keep track on the current closest squared distance

$$d_i^m = \min_{j \in \{1, \ldots, m\}} \|x_i - c_j\|^2 \tag{35}$$

of each element $x_i \in \boldsymbol{X}_n$ to the so far drawn centers $c_1, \ldots, c_m$ and sample the next center $c_{m+1}$ with probability $\propto (d_i^m)_{i=1}^n$ from $\boldsymbol{X}_n$. This procedure is repeated until the desired number of cluster centers is reached.

The advantage of k-means++ is that the initial centers are more diverse than if they were sampled at random from $\boldsymbol{X}_n$. However, in its standard form the centers initialized by k-means++ are themselves distributed

flatter than $\boldsymbol{X}_n$. And so, in the case of DC, we propose the following adjustment for a *distributional k-means*++:

We sample with probability $\propto \left(d_i^m p(x_i)^{2/d}\right)_{i=1}^n$ from $\boldsymbol{X}_n$, where $\boldsymbol{X}_n \sim p$.

**Symmetrized DC for molecules**  Since any symmetric molecule has multiple equivalent representations, care must be taken when measuring distances in DC. The key idea is to always compare the two configurations in its closest representation. Using the notation from Appendix B.4, let

$$d(z, z') = \min_{1 \leq s \leq \boldsymbol{s}} \|\Phi(z) - \Phi(\pi_s z')\|$$

be the symmetrized distance between two molecule representations. The symmetrized DC algorithm is then obtained by replacing all occurrences of $\|z - z'\|$ with $d(z, z')$ in the cluster assignments $I_c$, the objective $\mathrm{inertia}_{\mathrm{DC}}[\boldsymbol{c}|\boldsymbol{X}_n]$, the cluster updates $c^*$ and closest distances $d_i^m$ from Equations 33, 32, 34 and 35.

# D   Design choices of the sparse MoE model

In Sec. 3.1 we have made several design choices with computational feasibility in mind. We will discuss these summarized in this section.

**The gate model**  While in Sec. 3.1 we have chosen the gate $g_l \sim \mathcal{SGP}(\theta_{g_l})$ to be a GPR model, note that any choice of model with sufficient flexibility would have been possible. GPR features universal approximation properties, which makes it a favorable choice.

Furthermore, the gate should come with a small degree of freedom to prevent compared to the experts to prevent those from overfitting while training the MoE. For this reason, and the fact that we have no *ground truth* labels for the training of the gate anyhow, we choose our GPR-based gate to be sparse.

Finally, note that we share the set of gate inducing point locations $\boldsymbol{X}_G^*$ across all gate channels. While this is not necessary, it simplifies our method without costs as the MoE is rather insensitive with respect to the gate inducing point locations, as long as these are well-spread.

**The expert models**  While we made clear why we use GPR experts in our work, we left open in Sec. 3.1, whether these experts should be sparse or dense. Here, the deciding factor is plainly the amount of $n$ training samples that we have to deal with: When $n$ goes beyond a few thousands, we suggest switching to sparse GPR experts for computational reasons. Note that after training of the MoE, it is also possible to switch back to full GPR experts, if one aims for a high accuracy predictor. For the purpose of active learning, this is not necessary.

Similar to the gate, we share the inducing point locations $\boldsymbol{X}^*$ across all experts, which simplifies our model. In contrast to the gate situation, the MoE is sensitive to the choice of expert inducing point locations. Now, if we were to allow individual inducing point locations for each expert, it might happen that an elsewise locally underperforming expert works better than the remaining experts due to a lucky choice of its individual inducing points. Subsequently, this would result in a sub-optimal gate and, hence, ultimately in a wrong *superior training density* estimate.

For better generalization, if dense GPR experts are applied while training of the MoE, we will either have to rely on individual training sets for the experts and the gate, or we use *leave-one-out* expert responses on a shared training set.

In the sparse expert case, it is necessary to learn reasonable inducing values $\boldsymbol{\mu}^*$ prior to the actual learning procedure of the MoE to not get stuck in a spurious solution. Therefore, there should be a short pre-training phase of each individual expert.

In addition—whether or not the experts are sparse—the shared expert parameters $\mu_E, \lambda_E, \widehat{v}, \Sigma_E$ should be initialized reasonably. In this regard, we suggest to train a single, global expert model before the (pre-)training of the actual experts to obtain those initial parameter estimates for which we have no prior knowledge. If we assume isotropic bandwidths to be sufficient, we can simply set $\Sigma_E = \sigma_E \mathcal{I}_d$ and learn the scalar $\sigma_E > 0$ instead. Note that, from practice, the training of the MoE suffers tremendously from online changes of the expert bandwidths. Thus, we suggest to keep $\Sigma_E$ fixed after initialization.

Finally, note that, if we stick with sparse experts after training of the MoE, it can be beneficial for the prediction accuracy to re-train the MoE, where we keep the gate fixed. In this post-processing step, we would like to apply larger learning rates on the experts in order to escape local optima. However, larger learning rates also lead to underperforming intermediate steps, in which an actively trained gate might reject the best fitting expert at random—therefore pushing the gate towards a local optimum. Keeping the pre-trained gate fixed at this point prevents this undesired behavior.

**The inducing points** Recall that we have set the covariance $\boldsymbol{S}^* = 0$ of the inducing value distribution to zero, whereas it could have also been a diagonal or positive definite matrix. Playing around with this parameter, we have seen no significant improvement that would justify the considerable amount of additional model parameters from a computational point-of-view.

In our approach we suggest to keep the inducing point locations fixed, which is also for reasons of computational feasibility, but more importantly, adaptive inducing point locations come along with heavy prediction instabilities while training.

We found it necessary and sufficient to initialize the inducing point locations by state-of-the-art methods, as described in Appendix C.

**The MoE objective** While training of our MoE in Sec. 3.1.1, we added a penalty on small bandwidth choices. As described in (Lepski, 1991; Lepski & Spokoiny, 1997), the optimal bandwidth choice is the largest one which is capable of modelling the function. Now that we are able to model a comparably flat function by small bandwidths, as long as we have got enough training support, it can occur that, with no regularization, we choose a too small bandwidth for such a flat region. A too small bandwidth choice might cause overfitting. But even worse, in the subsequent active learning loop the flat region is falsely identified as complex, leading to more training queries in this location, which then allow for even smaller bandwidths to model this flat region. We will demonstrate this pathological behavior for the unregularized case on toy-data in Sec. 5.1.

**The gate noise 𝔰** Like already mentioned in Sec. 3.1.1, it is possible to tune 𝔰 in the training process:

**Remark 7.** *Shazeer et al. (2017) proposed to learn the 𝔰 parameter by adding a penalty term to the main objective that penalizes the imbalance of how likely training inputs are assigned to each expert: Let $\pi_b \in [0,1]^L$ be the expert assignment probabilities of $x_b$ and define $\pi_{\mathcal{B}} = \sum_{b \in \mathcal{B}} \pi_b$. Then they add a penalty $\mathbb{V}[\pi_{\mathcal{B}}]/[\mathbb{E}\,\pi_{\mathcal{B}}]^2$ to the objective, which is the* squared coefficient of variation—*a coefficient that accounts for the non-uniformity of a set of positive variables.*

We justify our simple heuristic to shrink 𝔰 in a static way as follows: Recall from Sec. 3.1 that 𝔰 prevents premature commitment to a spurious solution. When treating 𝔰 as a trainable parameter, it does not decay towards zero. Maintaining the noise then prevents the locally best performing experts from converging by randomly withholding training samples. For this reason, we find that 𝔰 behaves best when decaying towards zero as the training progresses.

# E Supplemental results on the Doppler experiment

## E.1 The single-scale GPR model

When training a single-scale GPR model on the Doppler dataset, the tuned bandwidth parameter will typically take an intermediate value, trying to compromise between more complex and simpler regions. This is reflected in the predictions in Fig. 13, where the single-scale GPR model suffers from the inhomogeneous structure, underfitting the complex region to the left while simultaneously overfitting the simple region to the right.

In Fig. 14 we compare the performance of our multi-scale MoE approach to the single-scale GPR model. The consistently inferior performance of the single-scale GPR model shows that the issue above persists even for large training sizes.

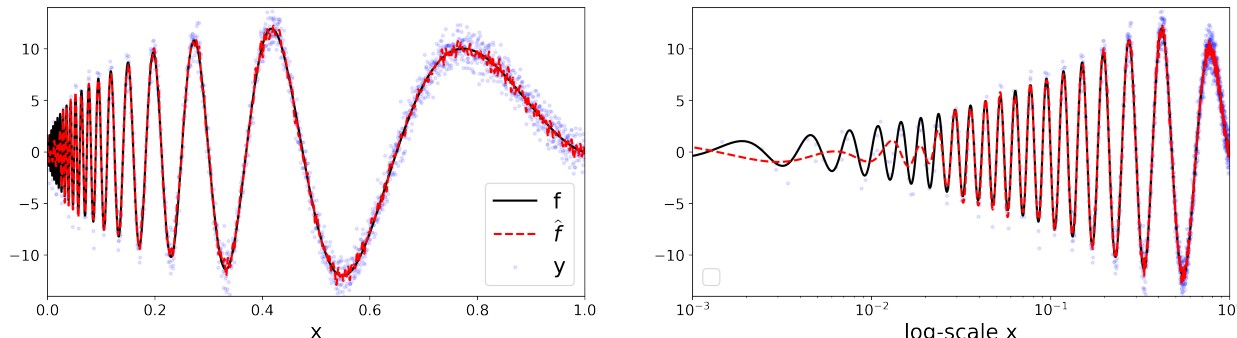

Figure 13: The Doppler experiment: An exemplary dataset and the predictions of a global GPR model, shown on natural x-scale (left) and on logarithmic x-scale (right).

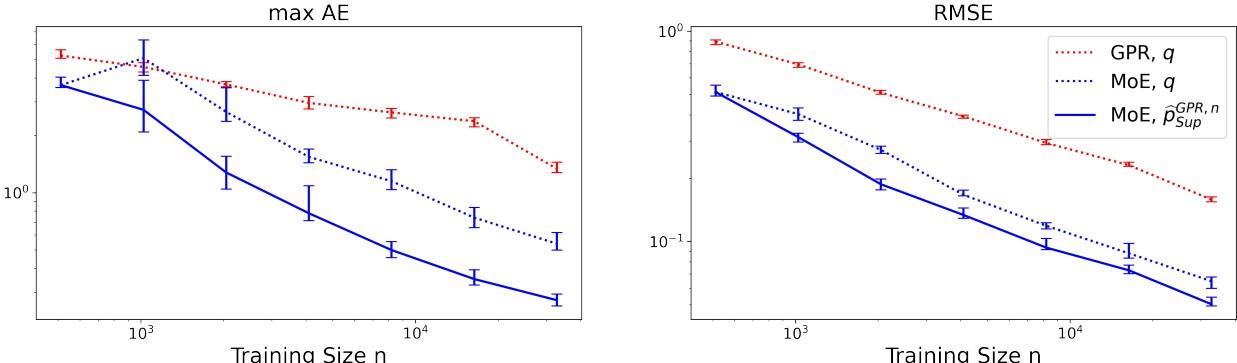

Figure 14: The Doppler experiment: The max AE (left) and the RMSE (right) of our proposed MoE model in comparison to a single-scale GPR model. The results are averaged over 20 repetitions.

## E.2 Necessity of the small bandwidth penalty

In Appendix D we discuss overfitting issues with too small local bandwidth estimates as a consequence of inadequate regularization of LOB. To address this issue, we have proposed to penalize such small bandwidth choices by $\vartheta_\sigma \mathrm{pen}_\sigma(\boldsymbol{X}_n, \boldsymbol{Y}_n, \mathcal{B}, w, \Theta)$ with the penalty term $\mathrm{pen}_\sigma$ from (13) and a scaling factor $\vartheta_\sigma \geq 0$.

Now, while the LOB estimate with $\vartheta_\sigma = 0.5$ (see Fig. 4) consistently behaves as expected, we show for comparison a typical LOB estimate in Fig. 15 that results from applying no regularization ($\vartheta_\sigma = 0$). By chance—here, the flat region of the Doppler function to the right—the trained model suffers from massive overfitting by too small LOB estimates. These falsely obtained small LOB estimates then lead to overestimation of LFC, which subsequently results in a detrimental oversampling of these locations by the active learning procedure.

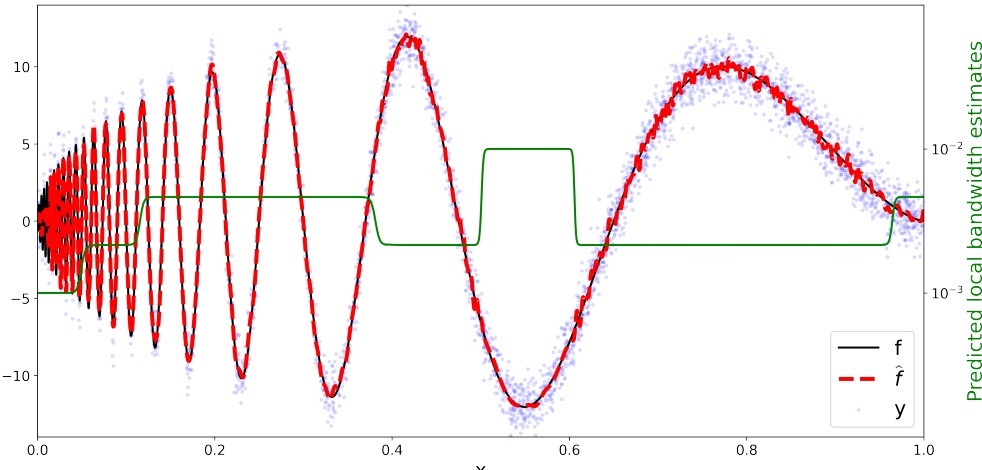

Figure 15: The Doppler experiment: An actively sampled dataset (top) with our MoE fit at $n = 2^{12}$ training samples without small bandwidth penalty ($\vartheta_\sigma = 0$), and the associated LOB estimate (bottom).

## F  The malonaldehyde MD simulation experiment under a uniform test distribution

In this scenario, we assume a uniform test density $q = \mathcal{U}(\mathcal{X})$. Accordingly, we weight the validation and test MSE by the importance weights $1/\widehat{p}_{\mathrm{MD}}(x_{\mathrm{val}})$ and $1/\widehat{p}_{\mathrm{MD}}(x_{\mathrm{test}})$. We draw the initial expert training set $\boldsymbol{X}_n$ of size $n = 2^9$ and the gate training set $\boldsymbol{X}_{n_G}^G$ via importance sampling from the remaining pool with weights $1/\widehat{p}_{\mathrm{MD}}(X_{\mathrm{MD}} \setminus (x_{\mathrm{val}} \cup x_{\mathrm{test}}))$. By this it is $\boldsymbol{X}_n \sim \mathcal{U}(\mathcal{X})$.

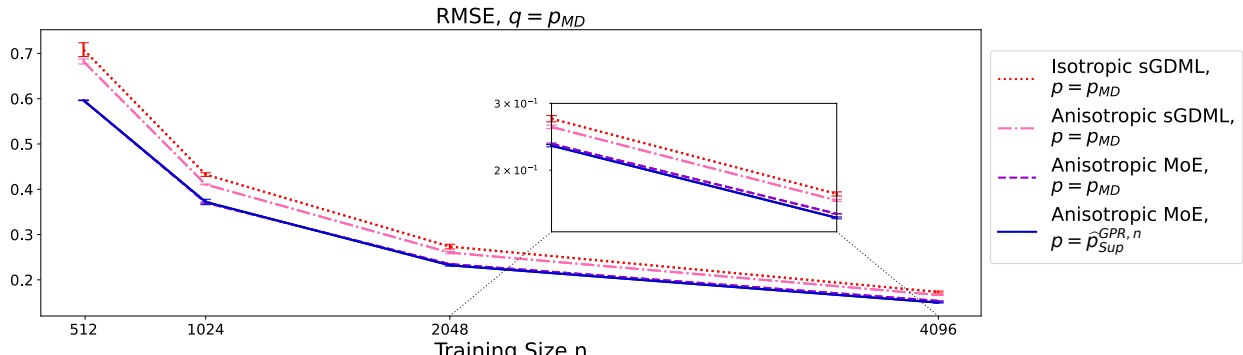

Figure 16: The RMSE under the uniform test distribution for different variants of sGDML and training distribution at varying training size: The performance is given for passive sampling, using the original isotropic sGDML (dotted), anisotropic sGDML (dash-dotted) and our MoE model with anisotropic sGDML experts (dashed), and for the MoE model, applying the proposed active learning framework (solid). The results are averaged over 2 repetitions.

In Fig. 17 we show the estimates to LOB, LFC and the *superior training density* under the pool test distribution, evaluated on the relaxed configurations of malonaldehyde. The LFC estimates in Fig. 17 confirm our expectation that the transition areas are more complex to model than the regions near the stable configurations. Subsequently, our active sampling scheme shifts sample mass away from the stable regimes in favor of the transition areas.

We have plotted the error curves of passive and active sampling schemes in Fig. 16. When estimating the relative sample size (19) that we require to achieve the same RMSE via active sampling compared to *random test sampling*, we obtain $\varrho(\widehat{f}_{\mathrm{MoE}}, \widehat{p}_{\mathrm{Sup}}^{\mathrm{GPR},n}) = 0.965 \pm 0.009$. This means that we save about 3.5% of samples with our active sampling scheme. With similar calculations, we save about 27%, when comparing the original sGDML approach with passive sampling to our MoE model with active sampling.

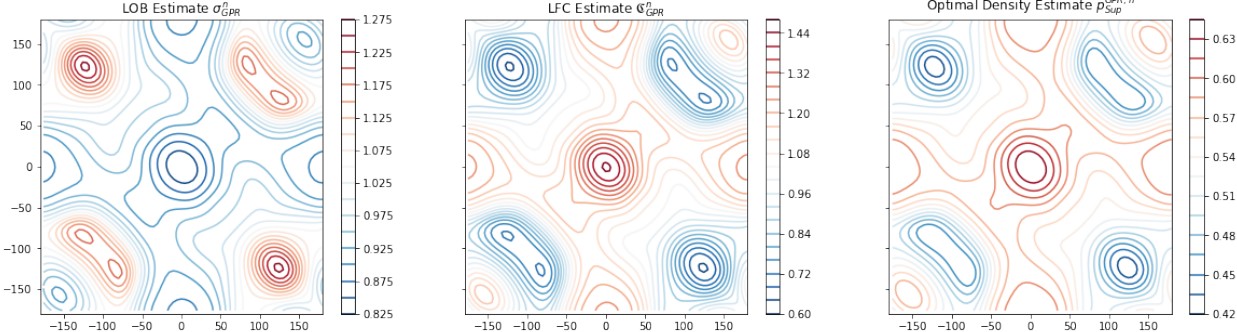

Figure 17: Estimates to LOB (left), LFC (middle) and the *superior training density* (right) under the pool test distribution $q = \mathcal{U}(\mathcal{X})$, evaluated at the relaxed malonaldehyde configurations, plotted with respect to the angles of the two aldehyde rotors of malonaldehyde.

