# OpenReview forum: "Local Function Complexity for Active Learning via Mixture of Gaussian Processes"
_TMLR — Rejected by TMLR_

### Review · Reviewer_vW7z · 2022-11-07

**Summary Of Contributions:**

The authors are interested in performing active learning for functions with varying spatial properties. They propose a gated, mixture of experts model, with Gaussian process experts with varying kernel bandwidth for this problem. Active learning is then performed by sampling from a density that the authors show is asymptotically optimal, in the sense of minimizing the expected mean squared error (where the expectation is both with respect to randomness in the data generating process and a distribution form which test points are drawn). This is essentially an extension of Panknin 2021 et al. who consider the case of local polynomial smoothing, to radial basis function kernels. The authors motivate this as allowing extensions to higher dimensional problems. The authors illustrate properties of their framework first on the doppler function, then on a force field reconstruction task.

**Audience:**

Yes

**Broader Impact Concerns:**

The paper does not raise significant ethical concerns that need further discussion.


**Claims And Evidence:**

Yes

**Requested Changes:**

## Important for acceptance
- The mathematical setup for the problem needs further detail in section 3. In particular, the authors define a local complexity function, and then use this to write down a density that they refer to as the optimal training density (inspired by Theorem 4). This seems to imply that a result like theorem 4 holds for this density. However, it doesn't appear that this result has been proven. Perhaps it follows easily from the results of Panknin et al, 2021 (I am not sure). Please include a formal argument for the optimality of the density. Alternatively, if it is not known whether or not this density is optimal in the sense of (asymptotically) minimizing MISE, please state this more clearly in the text, and perhaps refrain from referring to it as the optimal density as this could be misleading.
- At least a partial description of the training method used to fit the model to data should be included in the main text (perhaps deferring less important details to the supplement). The training procedure appears to be reasonably involved, and I think a bit of time in the text should be devoted to sketching it at least instead of deferring the entire description of it to the appendix.
- A more detailed description of how to choose hyperparameters stated at the end of section C.1 should be included. Note I do not necessarily mean the specific values used in experiments (although this is also appropriate to include), but a systematic way for choosing the many parameters such as number of inducing points, learning rate, number of experts in the model, etc. It appears to be a daunting list of things to tune, and without a clear description of how this can be done efficiently and systematically, I am concerned the method is not practical.

## Significant, but less important for acceptance
- In the case of sparse Gaussian process regression typically the mean and covariance matrix are computed in closed form, for example by minimizing the Kullback-Leibler divergence between the sparse Gaussian process and the posterior (e.g.~Seeger, Williams and Lawrence, 2003, and Titsias, 2009 "Variational Learning of Inducing Variables in Sparse Gaussian Regression"). The resulting mean basically has the interpretation as doing kernel ridge regression in a finite dimensional RKHS (determined by the inducing points). I would be curious to see a comparison between this sort of projection based approach and optimizing these parameters with respect to the objectives considered in this text. It seems it might be advantageous to reduce the number of parameters that need to be trained by computing $\mu$ (and potentially $S$) in closed-form.
- The authors should engage with existing literature on choosing inducing points for Gaussian process regression and landmark points for low-rank kernel ridge regression (essentially an equivalent problem), both of which are well-studied. To name a few reasonably early and well-known works looking at closely related problems: Smola and Scholkopf, 2000, "Sparse Greedy Matrix Approximation for machine learning" Seeger, Williams and Lawrence, 2003, "Fast forward selection to speed up Gaussian process regression", Fine and Scheinberg, 2001 "Efficient SVM Training Using Low-Rank Kernel Representations", Kumar, Mohri and Talwakar 2012 "Sampling Methods for the Nystrom method". Of course it isn't reasonable to discuss all of the methods considered in detail, but engagement with existing literature on selecting inducing points would benefit the paper.

## Minor Changes:
- Notation needs to be defined prior to or immediately following its use. In particular, It seems in equation 6 $p(x)$ and $\sigma^n_Q(x)$ are not defined in the main text, only in the supplement.
- The authors assume that the locally optimal bandwidth function is unique and exists. Some discussion of the regularity conditions required for this to exist in the setup considered should be added.
- Parentheses should likely be removed from section headers/paragraphs
- p13. "as opposed generating new data points on demand" -> "as opposed *to* generating new data points on demand"
- The authors claim that nearby inducing points lead to over-fitting. This claim should be supported.



**Strengths And Weaknesses:**

## Strengths
- The quality of writing in the paper is generally good.
- Similarly, many of the figures are both useful for supporting the writing and high-quality.
- The experiments seem to support the efficacy of the method.

## Weaknesses
- In places, the descriptions in the main text do not fully introduce notation, or skip important aspects of the proposed method. (See requested changes for details).
- The method has *many* parameters to tune. The authors discuss some balancing over-fitting and flexibility of the model. However, it isn't totally clear from the text how robust the findings are to the selection of these hyperparameters. I am concerned the method might be very difficult to use in practice because of the need to tune all of these hyperparameters.

---

> ### Author Response · Authors · 2022-12-06
> **Response to Review of Paper502 by Reviewer vW7z**
>
> Dear reviewer. Thank you for the suggestions to further improve our manuscript. We will revise our submission according to the comments below and provide an updated manuscript within the next days.
>
> # Essential for acceptance
> * Please include a formal argument for the optimality of the density. Alternatively, if it is not known whether or not this density is optimal in the sense of (asymptotically) minimizing MISE, please state this more clearly in the text, and perhaps refrain from referring to it as the optimal density as this could be misleading.
>
>  **Response**
>  This theorem is indeed not proven for GPR in our paper. Instead, we argue, that the model-agnosticity of the LPS-based Theorem 4 by Panknin et al. (2021), in combination with the conceptual similarity of LPS to GPR, justify the assumption that this theorem will apply for GPR to some extent. While we only claim an expected superior performance of this density for GPR in the text, we acknowledge that referring to the density as the *optimal training density* $p_{opt}^{GPR,n}$ can be misleading. We have decided to call it *superior training density* $p_{sup}^{GPR,n}$ for GPR in our revised manuscript.
>
>  * At least a partial description of the training method used to fit the model to data should be included in the main text
>
> **Response** We agree with the reviewer that our paper would benefit from more details about the employed training procedure in the main text. We have therefore moved the training objective and the training procedure from the appendix to Section 3.1 in our revised manuscript.
>
>  * A more detailed description of how to choose hyperparameters stated at the end of section C.1 should be included. Note I do not necessarily mean the specific values used in experiments (although this is also appropriate to include), but a systematic way for choosing the many parameters such as number of inducing points, learning rate, number of experts in the model, etc.
>
> **Response**
>  We have added a brief description in Section 3.1 on how to tune the essential hyperparameters on the initial training set in the revised manuscript.
>
> # Minor
> * In the case of sparse Gaussian process regression typically the mean and covariance matrix are computed in closed form. I would be curious to see a comparison between this sort of projection based approach and optimizing these parameters with respect to the objectives considered in this text. It seems it might be advantageous to reduce the number of parameters that need to be trained.
>
> **Response** We thank the reviewer for this suggestion. Indeed, our framework is modular with respect to the choice of GP experts. While treating different choices of expert models is out of scope of this work, we have addressed your point in the discussion and mention it as future work in the conclusion.
>
>  * The authors should engage with existing literature on choosing inducing points for Gaussian process regression and landmark points for low-rank kernel ridge regression (essentially an equivalent problem), both of which are well-studied. Of course it isn't reasonable to discuss all of the methods considered in detail, but engagement with existing literature on selecting inducing points would benefit the paper.
>
> **Response**
>  We have added a brief discussion on existing literature on inducing point choices in the related work section.
>
> # Suggestions
> * Notation needs to be defined prior to or immediately following its use. In particular, it seems in equation 6 $p$ and $\sigma_{Q}^{n}(x)$ are not defined in the main text.
>
>  **Response**
>  Thanks for pointing this out. We adapted the text before equation 5 and 6 accordingly.
>
>  * The authors assume that the locally optimal bandwidth function is unique and exists. Some discussion of the regularity conditions required for this to exist in the setup considered should be added.
>
>  **Response**
>  We adapted the text before equation 5 accordingly and added a footnote on this.
>
>  * Parentheses should likely be removed from section headers/paragraphs
>
> **Response**
>  OK
>
>  * p13. "as opposed generating new data points on demand" -> "as opposed to generating new data points on demand"
>
> **Response**
>  Thanks for the correction.
>
>  * The authors claim that nearby inducing points lead to over-fitting. This claim should be supported.
>
> **Response**
>  We would like to rephrase this claim.
>  The choice of inducing points can be interpreted as a nested active learning task at small sample size (number of inducing points). By choosing them diverse (with low-dispersion) we can achieve an acceptable representation of the input space for the subsequent inference at the smallest possible size. We have reformulated the paragraph in the revised Appendix C.2 accordingly.

---

> ### Author Response · Authors · 2022-12-15
> **Revision**
>
> Dear reviewer,
>
> We have uploaded a revised manuscript so as to address unclarities and concerns and to improve the submission according to your suggestions.
> The essential changes are highlighted by blue text.
>
> Sincerely,
> the authors

---

> > ### Comment · Reviewer_vW7z · 2022-12-19
> > **Definition 2**
> >
> > I don't understand what the $o(1)$ term in definition 2 is. Is all that is meant that the left and right hand sides are close? I guess if you mean that as $n$ tends to infinity the difference tends to $0$, this will be the case if the estimator is consistent under both training densities as both the "main terms" will tend to 0.
> >
> > Overall, if you are going to assess this empirically, I don't really see the reason to include the $o(1)$ term, and I worry it makes the definition unnecessarily complicated. I'd appreciate some clarification as to what is intended with the definition.

---

> > > ### Author Response · Authors · 2022-12-19
> > > **Clarification and correction with respect to Definition 2**
> > >
> > > Dear reviewer,
> > >
> > > The o(1) indeed symbolizes an error that goes to zero as $n\rightarrow \infty$.
> > > First of all there is a typo that needs for correction: Instead of a plain + o(1) it should have read
> > > $MISE\left(q,\widehat{f}| X_n\right) = MISE\left(q,\widehat{f}| X_{n'}'\right) (1 + o(1))$
> > > which means that both terms (left and right side) get closer to each other, even relative to their individual decay towards zero, as $n$ grows.
> > >
> > > Apologies for that. We will fix this error along with a few changes of $p_{Opt}$ to $p_{Sup}$ that we missed out.
> > >
> > > In this corrected, form we feel certain that this very formal definition of $\varrho$ is necessary. Also note that the other reviewers requested a formal definition in what sense our estimate $p_{Sup}$ is superior to naive training, which now is given in terms of $\varrho < 1$. Further note that with $\varrho$ we are able to rank different training densities in the sense that $\varrho(p) < \varrho(p')$ implies $p$ is better than $p'$ for arbitrary training densities $p,p'$.
> > >
> > > If the necessity of this definition is still unclear, let us try to elaborate as follows:
> > > We asymptotically have $MISE\left(q,\widehat{f}| X_n\right) = C_q n^{-\gamma}(1 + o(1))$ and $MISE\left(q,\widehat{f}| X_{n'}'\right) = C_p n^{-\gamma}(1 + o(1))$ for some constants $C_p, C_q > 0$ with respect to $n$.
> > > In order to be able to determine $\varrho$ empirically later on, we operate on this asymptotic form. That is, we try to equate $C_q n^{-\gamma}(1 + o(1)) = C_p {n'}^{-\gamma}(1 + o(1)) = C_p \varrho^{-\gamma} n^{-\gamma}(1 + o(1))$ such that $\varrho = (C_p/C_q)^{1/\gamma}(1 + o(1)) = (C_p/C_q)^{\beta}(1 + o(1))$ for $\beta = 1/\gamma$.
> > >
> > > Empirically, we can estimate the ratio $C_p / C_q$ from finite data. But we needed a proper definition of $\varrho$ in the beginning to understand how to translate the observed MISE's under varying training distributions to the relative required sample sizes (when applying p instead of q for training).

---

### Review · Reviewer_hJUx · 2022-11-29

**Summary Of Contributions:**

The paper proposes a Gaussian process-based model for estimating locally optimal bandwidth in regression settings where the true function changes complexity, e.g., lengthscales, over its domain and/or is observed with heteroscedastic noise. Given a radial basis function-based predictor, the optimal bandwidth(s) at a point is defined as the lengthscale value(s) which minimizes mean-squared error at that point (the expectation is taken over the additive heteroscedastic noise). The paper describes a mixture of expert GP models that can be used to estimate local bandwidths after which, through a connection with local function complexity described in previous work, a sampling density for active learning can be established. A set of experiments on a 1-D toy example highlights some benefits of the proposed approach. An additional experiment in quantum chemistry demonstrates sample size reduction.

**Audience:**

Yes

**Claims And Evidence:**

No

**Requested Changes:**

* The exposition switches from assuming heteroscedastic noise prior to section 3.2 to assuming homoscedastic noise thereafter. Moreover, it doesn't seem that the experiments model heteroscedastic noise. Why not assume homoscedastic noise throughout? Especially, since the optimal sampling density relies on the pointwise variance of the additive noise, it doesn't seem useful to consider heteroscedastic noise. Similarly, the true function smoothness, denoted by \alpha, is needed for sampling in the active learning framework described in section 3.3. However, it is assumed in the experiments that the true function is infinitely smooth so that the sampling density is described instead by (12). Why not just assume \alpha = \infty throughout? Especially, given that the paper does not provide guidance on how this smoothness factor would be estimated from data.

* There are a number of parameters set to certain values as documented in the experimental and supplemental sections. If the proposed method were to be utilized within a novel active learning setting, how would one set these values?

* The motivation for the use of GP-based estimates for optimal bandwidths is motivated by the observation that GPs don't suffer in high dimensions as does the previous estimate described in Panknin et al. (2021). However, there is no demonstration of this in the 1- and 2-D experiments. Why wasn't the LPS method of Panknin et al. (2021) applied there as a competitor?

* Please provide a succinct statement on what the "optimal" of optimal training density refers to.

**Strengths And Weaknesses:**

An effort has obviously been made to guide the development with sound technical considerations. For instance, the jump to the refined bandwidth estimates from the previous local polynomial smoothing-based estimates seems reasonable.

There are many details involved in the proposed approach which makes me wonder whether it is broadly applicable or whether it is solving very specific problems. Also, it is not clear how it fares generally against other active learning approaches. I'm not an expert but the provided experimental evidence seems weak - are there not established benchmarks for active learning methods for which the proposed method could be evaluated? Is it truly the case that the only competitor is the described GP-based uncertainty approach? As much as I liked the development being guided by what appear to be sound technical considerations, there are also assumptions that are not obviously satisfied in practical settings (or cannot be obviously verified to be satisfied). One example is the assumed smoothness level of the true function which is necessary in the definition of the optimal training density. A simpler and more self-contained exposition may also be helpful in conveying benefits.

---

> ### Author Response · Authors · 2022-12-06
> **Response to Review of Paper502 by Reviewer hJUx**
>
> Dear reviewer. Thank you for the suggestions to further improve our manuscript. We will revise our submission according to the comments below and provide an updated manuscript within the next days.
> # Requested Changes
>  * It is not clear how it fares generally against other active learning approaches. I'm not an expert but the provided experimental evidence seems weak - are there not established benchmarks for active learning methods for which the proposed method could be evaluated? Is it truly the case that the only competitor is the described GP-based uncertainty approach?
>
>  **Response**
>  Our work considers the domain of model-agnostic active learning at large training sizes. Here, we delimit us from active learning solutions and benchmark datasets that are concerned with small sample sizes or parametric models.
>  We extend the previous work by Panknin et al. (2021), who have already compared favorably to other model-agnostic active learning approaches regarding performance and model-agnosticity. Hence, we imply this favorable behavior of our approach by transitivity.
>  To clarify this issue, we have revised the discussion on related AL approaches in Section 4.
>
>  * The exposition switches from assuming heteroscedastic noise prior to section 3.2 to assuming homoscedastic noise thereafter. Moreover, it doesn't seem that the experiments model heteroscedastic noise. Why not assume homoscedastic noise throughout? Especially, since the optimal sampling density relies on the pointwise variance of the additive noise, it doesn't seem useful to consider heteroscedastic noise. Similarly, the true function smoothness, denoted by $\alpha$, is needed for sampling in the active learning framework described in section 3.3. However, it is assumed in the experiments that the true function is infinitely smooth so that the sampling density is described instead by (12). Why not just assume $\alpha = \infty$ throughout? Especially, given that the paper does not provide guidance on how this smoothness factor would be estimated from data.
>
>  **Response**
>  From the theoretical point-of-view, we obtain our result simultaneously for all smoothness levels $\alpha$, which is a contribution on its own. This is why we state this result in its readily available most general form, even though we present no experimental survey about this parameter of the learning problem.
>  To delimit the scope of this work we tried to avoid heteroscedastic treatment in the experiments, noting that we presented the readily available theory for the same reasons as for the function smoothness.
>  However, the heteroscedastic treatment is well elaborated in the domain of Gaussian processes, and we are able to add an experiment on the Doppler function with heteroscedastic noise without exceeding the current length of our manuscript by a lot.
>
>  * There are a number of parameters set to certain values as documented in the experimental and supplemental sections. If the proposed method were to be utilized within a novel active learning setting, how would one set these values?
>
>  **Response**
>  We have added a brief description in Section 3.1 on how to tune the essential hyperparameters on the initial training set in the revised manuscript.
>
>  * The motivation for the use of GP-based estimates for optimal bandwidths is motivated by the observation that GPs don't suffer in high dimensions as does the previous estimate described in Panknin et al. (2021). However, there is no demonstration of this in the 1- and 2-D experiments. Why wasn't the LPS method of Panknin et al. (2021) applied there as a competitor?
>
>  **Response**
>  We have compared to the LPS method of Panknin et al. (2021) for the 1-D synthetic experiment (see the paragraph `Comparing the active learning framework in the LPS and GPR domain' in Section 5.1). Unfortunately, we denoted the LPS model of order 3 as LCS (for local cubic smoothing) in Figure 7, without a proper definition of this term, which we have corrected for. Regarding the force field reconstruction experiment in Section 5.2, the inputs are 27-dimensional of which we plotted the most descriptive 2-D projection for visualization purpose only. All calculations are performed in the high-dimensional space.
>  We have revised the experimental description to stress this circumstance and clarify our experimental setup. We reiterate, that our experiment does indeed demonstrate the capability of our GP-based approach to deal with high dimensions. Here, a comparison with the related LPS-based method is intractable.
>
>  * Please provide a succinct statement on what the 'optimal' of optimal training density refers to.
>
>  **Response**
>  While we only claim an expected superior performance of this density for GPR in the text, we acknowledge that referring to the density as the *optimal training density* $p_{opt}^{GPR,n}$ can be misleading. We have decided to call it *superior training density* $p_{sup}^{GPR,n}$ for GPR in our revised manuscript.

---

> ### Author Response · Authors · 2022-12-15
> **Revision**
>
> Dear reviewer,
>
> We have uploaded a revised manuscript so as to address unclarities and concerns and to improve the submission according to your suggestions.
> The essential changes are highlighted by blue text.
>
> Sincerely,
> the authors

---

### Review · Reviewer_LN9V · 2022-11-29

**Summary Of Contributions:**

Summary

This paper presents an active learning approach for a mixture of experts model over Gaussian process regression models with varying bandwidths. The authors leverage existing results on LPS to construct an experimental design for their model. They show improved empirical performance on a synthetic dataset and a chemistry motivated dataset.



**Audience:**

Yes

**Claims And Evidence:**

No

**Requested Changes:**

Please see above. Overall I would like to see a clearer presentation.

**Strengths And Weaknesses:**


Overall Impressions

Overall I found the paper interesting. Adaptive experimental design for Gaussian processes that extends beyond naive sampling which greedily optimize acquisition functions is a nascent and quickly developing area.

My main concern with the paper is that I found it poorly structured and difficult to read. Here are several specific comments:

1) I think the authors assumed a bit too much background knowledge from the reader on the past work by Panknin et al ’21. I had to dive fairly deeply into the appendix to even understand the basic setup, and I think the authors could have done a better job guiding the reader through the contributions of this past work.

2) There were many spots in the paper where I did not feel that things were appropriately defined. For example, the noise variance $\sigma^2_{\epsilon}$ on page 6. Is it the same as the noise variance $\nu(x)$? Also, the model itself was a bit poorly specified - for example I could not tell what the loss function was. It seemed to be given in appendix C, but I still think that details like this are important enough to include in the main text.

3) Returning to the main experimental design in equation (5), I was confused by the computation of the optimal sampling distribution p. It seems that $p _{opt}$ is what we are trying to compute, but it also seems to arise in the definition of $\mathfrak{C}$ - the local function complexity. This is a bit circular? Furthermore, I didn’t fully understand what “optimal” meant in this setting. Optimal with respect to what? Looking in the appendix didn’t really shine any additional light on this - Theorem 1 was somewhat vague on this as well.

4) As a bigger question, why should I expect the design motivated through the LFC to be the optimal thing to do in the case of this mixture of experts? From that perspective, presumably I could use any other model class as well that can appropriately approximate

5) By the time all the details on LPS and LFC and GPR and MOE are given, I have lost a bit of the original motivation. Why is this a good model to capture heteroscedasticity as motivated by the abstract?

Empirical Results

The empirical section seemed extensive, but again, I found it difficult to parse. For example, I didn’t follow the discussion on the bottom of page 9. Why should the inducing points (presumably those used to train the GP’s from the points that are actively chosen) follow the LFC? This lack of understanding probably arises because very little explanation of the LFC has been given. Also, in Figure 6, what is the training distribution p - I thought we were sampling from the optimal distribution $p_{opt}?
Other minor comments:

a) In the definition of the Doppler function, maybe don’t use p(y|x) since p means something else.

b) What is absolute error (AE)? It seems to not be defined? Also is LCS in Figure 7 supposed to be LPS?

c) Finally, I am a bit worried about the lack of comparisons to other methodologies. It seems that RMSE is compared for MOE and LPS, but for no other methods. There are a variety of methods out there for active regression. Why are there no other baselines?

The force field example seems like a very interesting application, but I similarly failed to understand it due to a lack of detail and explanation.

---

> ### Author Response · Authors · 2022-12-06
> **Response to Review of Paper502 by Reviewer LN9V**
>
> Dear reviewer. Thank you for the suggestions to further improve our manuscript. We will revise our submission according to the comments below and provide an updated manuscript within the next days.
> # Main concerns
> 1.
>
> **Response**
> If we understand correctly, the reviewer does not criticize the summary of their theory and methodology but an insufficient presentation of their contributions and implications. We have added the respective details to the end of Section 2.2.
>
> 2.
>
> **Response**
> While v is the local noise variance function of the regression task, $\sigma_\varepsilon^2$ is the noise variance parameter of a GP model in more general. When considering an expert of our MoE model, $\sigma_\varepsilon^2$ shall indeed be an estimate to v. However, when considering a gate-channel, it has a different meaning, since the gate does not model the target function. We added a note at the place in question to clarify this difference.
>
> We agree with the reviewer that our paper would benefit from more details about the employed training procedure in the main text. We have therefore moved the training objective and the training procedure from the appendix to Section 3.1 in our revised manuscript.
>
> 3.
>
> **Response**
> In fact, the sampling process is circular, as depicted in Figure 2, but there is no circular use of p. However, we acknowledge that some details were missing to avoid this confusion, which we have added in Section 3.
>
> By 'optimal' we mean that a training sample which is distributed according to $p_{opt}$ will (asymptotically) solve the active learning task in Equation (3). However, this optimality is proven only for LPS, and we rely on the model-agnosticity of this result to obtain a superior training density (with respect to naive sampling according to the test density). We acknowledge that referring to the density as the *optimal training density* $p_{opt}^{GPR,n}$ can be misleading. We have decided to call it *superior training density* $p_{sup}^{GPR,n}$ for GPR in our revised manuscript.
>
> 4.
>
> **Response**
> The reviewers implication is correct. The 'why' question is probably related to the insufficient presentation of the contributions and implications of Panknin et al. (2021) (see the first response).
>
> 5.
>
> **Response**
> In the revised manuscript we make sure to have a note on how the above terms are related to our goal after their definition.
>
> To delimit the scope of this work we tried to avoid heteroscedastic treatment in the experiments, while presenting the theory in its most general, readily available form.
> However, the heteroscedastic treatment is well elaborated in the domain of Gaussian processes, and we are able to add an experiment on the Doppler function with heteroscedastic noise without exceeding the current length of our manuscript by a lot.
>
> # Empirical Results
> * Why should the inducing points follow the LFC? This lack of understanding probably arises because very little explanation of the LFC has been given.
>
> **Response**
> In the revised manuscript we give more details in Section 2.2 about LFC and other properties.
>
> * Also, in Figure 6, what is the training distribution p - I thought we were sampling from $p_{opt}$?
>
> **Response**
> For comparison we run experiments with training density $p=p_{opt}$, but also the naive baseline of $p=q$ (following the test distribution). Based on the applied p, we can furthermore choose inducing points according to different strategies. In the revised manuscript, Figure 3 will show the most naive inducing points (following p under the naive training density p=q) in comparison to the most sophisticated choice.
>
> * In the definition of the Doppler function, maybe don’t use p(y|x) since p means something else.
>
> **Response**
> OK
>
> * What is absolute error (AE)? It seems to not be defined?
>
> **Response**
> By AE we mean the absolute value $|f(x)-\widehat{f}(x)|$ between the target function and the prediction evaluated in x. We added the definition at its first occurrence.
>
> * Also is LCS in Figure 7 supposed to be LPS?
>
> **Response**
> Yes. We have corrected for this.
>
> I am a bit worried about the lack of comparisons to other methodologies.
>
> **Response**
> Our work considers the domain of model-agnostic active learning at large training sizes. Here, we delimit us from active learning solutions and benchmark datasets that are concerned with small sample sizes or parametric models.
> We extend the previous work by Panknin et al. (2021), who have already compared favorably to other model-agnostic active learning approaches regarding performance and model-agnosticity. Hence, we imply this favorable behavior of our approach by transitivity.
> To clarify this issue, we have revised the discussion on related AL approaches in Section 4.
>
> * The force field example seems like a very interesting application, but I similarly failed to understand it due to a lack of detail and explanation.
>
> **Response**
> We have revised the experimental description to clarify our experimental setup.

---

> ### Author Response · Authors · 2022-12-15
> **Revision**
>
> Dear reviewer,
>
> We have uploaded a revised manuscript so as to address unclarities and concerns and to improve the submission according to your suggestions.
> The essential changes are highlighted by blue text.
>
> Sincerely,
> the authors

---

### Decision · Action_Editors · 2023-01-19

**Recommendation:** Reject

**Comment:**

The manuscript presents contributions that could be of interest to TMLR audiences. However, clarity and empirical support of the claims are significantly lacking. Therefore, the manuscript in its current form is not ready to be published in TMLR. The authors are encouraged to make a significant revision that will improve the issues raised by the reviewers, and resubmit to TMLR.

**Audience:**

The paper does include contributions that could be of interest to some in TMLR's audience.

**Claims And Evidence:**

The claims and evidence provided in the paper are insufficient:
The definition of the setup is incomplete and lacks clarity. The experimental evidence is missing ablation experiments as well as appropriate baselines. In addition, the paper is hard to read, and there remain numerous clarity issues even in the revised version that the authors uploaded.  Lastly, a better explanation of the central ideas of the paper is required.